

# Enhanced Continual Learning of Vision-Language Models with Model Fusion

**Haoyuan Gao**[1,*]**, Zicong Zhang**[1,*]**, Yuqi Wei**[1]**, Linglan Zhao**[1]

**Guilin Li**[4]**, Yexin Li**[3]**, Bo Wang**[4]**, Linghe Kong**[1]**, Weiran Huang**[1,2,†]

[1] Shanghai Jiao Tong University    [2] Shanghai Innovation Institute

[3] State Key Laboratory of General Artificial Intelligence, BIGAI    [4] Tencent

## Abstract

Vision-Language Models (VLMs) represent a significant breakthrough in artificial intelligence by integrating visual and textual modalities to achieve impressive zero-shot capabilities. However, VLMs are susceptible to catastrophic forgetting when sequentially fine-tuned on multiple downstream tasks. Existing continual learning methods for VLMs face various limitations, often relying on additional reference datasets, compromising zero-shot performance, or being restricted to parameter-efficient fine-tuning scenarios. In this paper, we propose a novel Continual Decoupling-Unifying (ConDU) approach that pioneers the use of model fusion for continual learning in VLMs. Specifically, ConDU maintains a unified model along with task triggers and prototype sets, employing an iterative process of decoupling task experts for previous tasks and unifying them with the task expert for the newly learned task. Additionally, we introduce an inference strategy for zero-shot scenarios by aggregating predictions from multiple decoupled task experts. Extensive experiments on the MTIL benchmark show that ConDU achieves up to a 2% improvement in average performance across all seen tasks compared to state-of-the-art baselines, while also enhancing zero-shot capabilities relative to the original VLM. Our code is available at https://github.com/zhangzicong518/ConDU.

## 1 Introduction

Artificial Neural Networks (ANNs) often suffer a significant performance drop on earlier tasks when learning sequentially. This issue, known as catastrophic forgetting (McCloskey & Cohen, 1989; Ramasesh et al., 2020), limits the adaptability of ANNs in dynamic environments. To overcome this challenge, continual learning (also referred to as lifelong learning) (Zenke et al., 2017; Kirkpatrick et al., 2017; Verwimp et al., 2023; Shi et al., 2024) has been developed. This paradigm aims to enable machine learning models to acquire new knowledge over time while preserving previously learned information, thus mimicking the adaptability of the human brain.

Recently, Vision-Language Models (VLMs) such as CLIP (Radford et al., 2021) have made a major breakthrough in artificial intelligence by integrating visual and textual modalities to achieve impressive zero-shot capabilities. However, despite their demonstrated success (Shen et al., 2021; Zhao et al., 2023b; Fan et al., 2024), VLMs remain susceptible to catastrophic forgetting when fine-tuned for multiple downstream tasks. Conventional continual learning approaches are insufficient for VLM fine-tuning, as they struggle to maintain the crucial zero-shot capabilities that make these models valuable (Zheng et al., 2023).

In contrast to the extensive research on conventional continual learning, relatively few methods (Zheng et al., 2023; Yu et al., 2025; Park, 2024; Xu et al., 2024) have been proposed for continual learning of VLMs. Some methods, such as (Zheng et al., 2023) and (Yu et al., 2025), require ad-

---

[*]Haoyuan (gaohaoyuan@sjtu.edu.cn) and Zicong contributed equally to this work. This work was conducted at MIFA Lab (members from SJTU & SII).

[†]Corresponding author.

ditional reference datasets for distillation from pre-trained models, and their performance is highly sensitive to the choice of the dataset (Zheng et al., 2023). Moreover, these methods require careful tuning of multiple handcrafted hyperparameters to balance different optimization objectives: mitigating catastrophic forgetting, preserving zero-shot capabilities, and optimizing performance on the current task. Alternative methods (Yu et al., 2024; Park, 2024; Xu et al., 2024) focus exclusively on parameter-efficient fine-tuning (Ding et al., 2023) employing modules such as adapters or LoRA (Hu et al., 2021), but struggle to adapt to full fine-tuning scenarios.

To overcome these limitations, we propose the *Continual Decoupling-Unifying* (ConDU), a novel continual learning approach for VLMs that is the first to introduce model fusion for this purpose. Model fusion (Ilharco et al., 2022; Yang et al., 2023; Huang et al., 2024) is a technique that combines multiple models into a single unified model without requiring access to the original training data. This property is particularly well-suited for the sequential learning scenario, as it allows one to maintain a single unified model that can be decoupled into multiple task experts to handle different tasks. However, a direct application of model fusion (iteratively merging new task experts into a single unified model) is unsuitable for continual learning as it causes severe performance degradation. Therefore, we carefully designed decoupling-unifying framework that avoids this issue by incorporating new task experts at the individual model level. Furthermore, ConDU is inherently compatible with both parameter-efficient and full fine-tuning paradigms, offering a flexible solution for diverse continual learning scenarios.

ConDU maintains a unified delta model and a set of task triggers throughout the continual learning process. ConDU handles each new task by fine-tuning the pre-trained VLM to obtain its task expert, decoupling to obtain past task experts via task triggers, and unifying all task experts into an updated unified model with new task triggers. We remark that the decoupling and unifying procedures introduced in ConDU are *training-free*, and thus their running time is much shorter than the time required for model fine-tuning. Moreover, compared to previous continual learning methods for VLMs mentioned earlier, ConDU eliminates the need for adjusting trade-off hyperparameters, incorporating reference datasets, and maintaining replay exemplars.

After the above continual learning process, our method supports multiple inference scenarios. If the test sample belongs to a previously seen task and its task ID is known, we can directly reconstruct the corresponding task expert and use it for prediction. When the task ID is unknown or the test sample comes from an unseen task (*i.e.*, the zero-shot scenario), we can instead reconstruct multiple task experts relevant to the test sample's domain and make a prediction by aggregating their results. Evaluated on widely used benchmarks across diverse settings, including Multi-domain Task Incremental Learning (MTIL), few-shot MTIL, and task-agnostic MTIL, ConDU achieves up to a 2% improvement in average performance across all seen tasks compared to state-of-the-art baselines, demonstrating the effectiveness of incorporating model fusion. Moreover, ConDU exhibits strong zero-shot capabilities of VLMs, outperforming the original pre-trained VLM and other state-of-the-art continual learning methods.

The contributions of this work can be summarized as follows:

- We introduce model fusion into continual learning for VLMs and propose a novel Continual Decoupling-Unifying (ConDU) framework, which is compatible with both parameter-efficient and full fine-tuning paradigms.
- We propose aggregating the predictions of multiple decoupled models for zero-shot scenarios.
- Through extensive experiments, we demonstrate that ConDU effectively learns new knowledge while preserving previously acquired knowledge and enhancing zero-shot capabilities.

## 2 RELATED WORK

**Continual Learning for VLMs.** Conventional continual learning has been extensively studied, including architecture-based methods (Rusu et al., 2016; Mallya & Lazebnik, 2018; Serra et al., 2018), replay-based methods (Riemer et al., 2018; Buzzega et al., 2020; Boschini et al., 2022; Gao & Liu, 2023; Kim et al., 2024), and regularization-based methods (Kirkpatrick et al., 2017; Zenke et al., 2017; Li & Hoiem, 2017; Zhao et al., 2023a; Lu & Sun, 2024). However, these methods cannot be directly applied to recently developed Vision-Language Models (VLMs), as they struggle to maintain the crucial zero-shot capabilities (Zheng et al., 2023).

Recently, continual learning methods specifically designed for VLMs have been introduced. These methods can be broadly classified into parameter-efficient fine-tuning based approaches (Wang et al., 2022; Yu et al., 2024; Park, 2024; Li et al., 2024; Xu et al., 2024) and distillation-based methods (Ding et al., 2022; Zheng et al., 2023; Yu et al., 2025). However, these methods either require reference datasets or the careful adjustment of trade-off hyperparameters, or they are not suitable for full fine-tuning. In contrast, our method eliminates these drawbacks by introducing model fusion.

**Model Fusion.** Model fusion combines multiple models into a single unified model, retaining the strengths of its constituent models without requiring additional training data. Fisher Merging (Matena & Raffel, 2022) and RegMean (Jin et al., 2022) use Fisher information matrices (Fisher, 1922) and inner-product matrices (Jin et al., 2022), respectively, to compute fusion coefficients for weighted model fusion. Task Arithmetic (Ilharco et al., 2022) introduces a fusion technique that combines models by summing delta models, where a delta model is defined as the difference between the parameters of a fine-tuned model and its pre-trained counterpart. Other approaches, such as TIES Merging (Yadav et al., 2024), Ada Merging (Yang et al., 2023), DARE (Yu et al., 2023), and EMR Merging (Huang et al., 2024), focus on enhancing delta model-based fusion in various ways.

## 3 PROBLEM FORMULATION

In this paper, we focus on continual learning for Vision-Language Models (VLMs). Given a pre-trained VLM (*e.g.*, CLIP (Radford et al., 2021)), a sequence of $T$ tasks arrives incrementally, where each task $t$ is associated with a training dataset $\mathcal{D}^t$. These tasks may involve distinct classes, different domains, or exhibit significant variation in sample sizes. After seeing each task, the VLM can be updated with access to a limited memory storing essential information (*e.g.*, selected past data or parameters). The goal is to develop a method that incrementally updates the VLM while achieving high performance on all previously encountered tasks and retaining its zero-shot capabilities for unseen tasks. Additionally, we aim for the proposed continual learning method to support both parameter-efficient fine-tuning (*e.g.*, LoRA or adapters) and full fine-tuning.

Under the continual learning setting, the system is permitted to retain only a single VLM throughout training. Yet, if multiple models were allowed, one could simply fine-tune a separate model from the pre-trained VLM for each task and choose the corresponding model at test time whenever the task identity is known. In addition, a defining property of VLMs is their zero-shot ability, which ideally should be preserved—or even improved—after continual learning. For test samples from unseen tasks (the zero-shot case), using several specialized models fine-tuned on different domains and aggregating their predictions would naturally outperform relying on a single VLM.

Motivated by this observation, if the shared components across these individually fine-tuned models could be extracted and merged into one maintained VLM, while the task-specific differences are stored in limited memory, then a single main VLM plus small auxiliary memory could effectively mimic the behavior of multiple task-specialized models. Moreover, this idea is inherently compatible with both parameter-efficient fine-tuning and full fine-tuning approaches.

## 4 CONDU: CONTINUAL DECOUPLING-UNIFYING

We propose *Continual Decoupling-Unifying* (ConDU), a novel continual learning approach for VLMs that leverages model fusion. Figure 1 shows the overall framework of ConDU. ConDU maintains a unified model, a set of task triggers, and a series of prototype sets throughout the continual learning process. Our framework includes five modules. We will introduce three modules for training in Section 4.1 and two modules for inference in Section 4.2.

### 4.1 DELTA MODELS CONTINUALLY FUSION AT TRAINING STAGE

At each session $t$ of the continual learning process, ConDU implements three steps: Tuning Individually, Decoupling Unified Model, and Unifying Models. The time spent on Decoupling Unified Model and Unifying Models is nearly 1% of Tuning Individually (see Appendix I for detailed analysis). Since the process of Decoupling Unified Model relies on the task triggers produced during

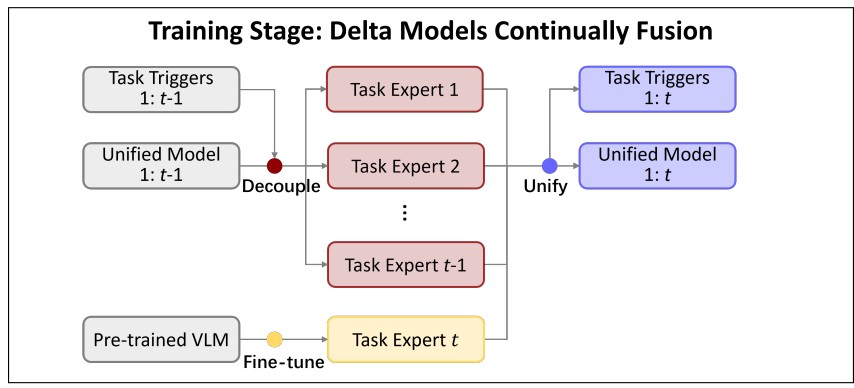

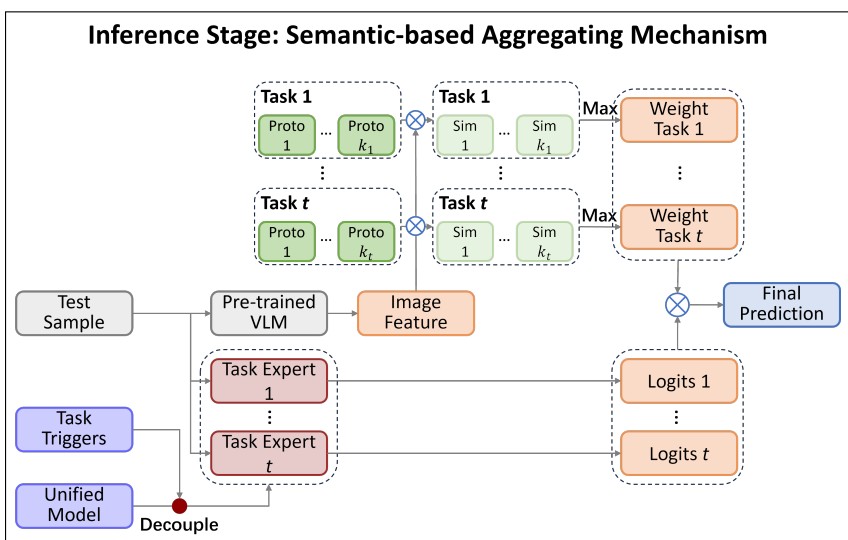

Figure 1: Overall framework of the proposed method. This framework includes designs for both the training stage and the inference stage. The upper part of the figure corresponds to the training stage of session $t$, with the relevant components detailed in Section 4.1. The colored points "unify" and "decouple" illustrate the corresponding operations, which are explained in Figure 2a and Figure 2b, respectively. During the training stage, ConDU handles each new task by fine-tuning the pre-trained VLM to obtain its task expert, decoupling to obtain past task experts via task triggers, and unifying all task experts into an updated unified model with new task triggers. The lower part corresponds to the inference stage after session $t$, with its components detailed in Section 4.2. During the inference stage, ConDU calculate the cosine similarity between the image feature of the test sample and prototypes of each category in the feature space of pre-trained VLM, then choose the maximum similarity in each task as the weight of the corresponding task expert.

Unifying Models, we first introduce the process of Unifying Models before detailing process of Decoupling Unified Model.

**Tuning Individually.** We denote a VLM as $f(\cdot; \theta)$, where $\theta$ represents only the learnable parameters of the VLM, excluding the frozen parameters for clarity. At session $t$, by fine-tuning the pre-trained VLM $\theta^0$ on task $t$, we obtain a task expert $\theta^t$. We defined delta model $t$, the parameter offsets of task expert $t$ relative to the pre-trained VLM, as $\delta^t = \theta^t - \theta^0$. We will unify delta models instead of directly unifying task experts following the setting of advanced model fusion methods (Ilharco et al., 2022; Yadav et al., 2024; Huang et al., 2024).

**Unifying Models.** The process of Unifying Models is illustrated in Figure 2a. When task $t$ arrives, we first decouple the current unified delta model $\delta^{1:t-1}$ to obtain the approximation of $\delta^i$ denoted as $\tilde{\delta}^i$ for $i = 1, 2, \ldots, t - 1$ (this process will be introduced in the paragraph of Decoupling Unified Model). Then let $\delta^i \leftarrow \tilde{\delta}^i, i = 1, 2, \ldots, t - 1$, the calculation of unified delta model is

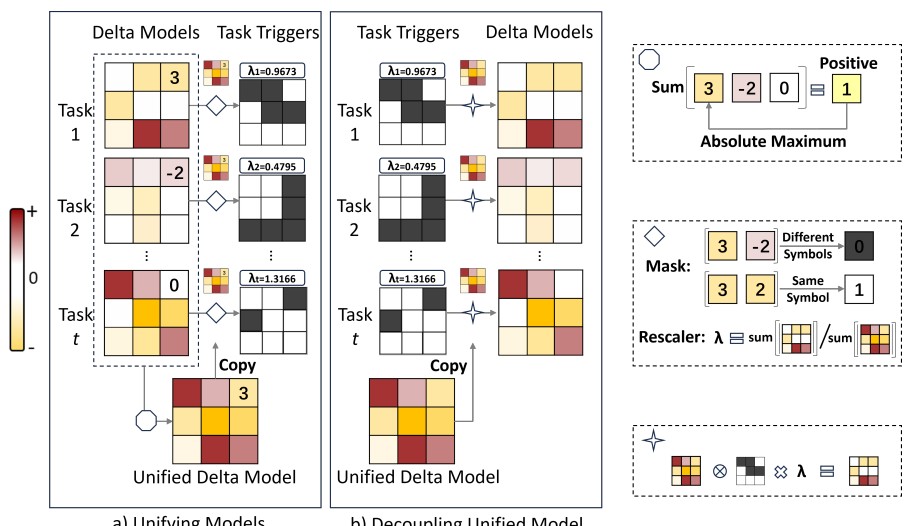

Figure 2: The process of Unifying Models (a) and Decoupling Unified Model (b) is transformed to unifying delta models (a) and decoupling unified delta model (b), respectively. Unified Model = Unified Delta Model + Pre-trained VLM. Task Expert $i$ = Delta Model $i$ + Pre-trained VLM. a) When unifying delta models, the unified model is obtained by an election process. Each task's task trigger is calculated according to the difference between the delta model and the unified delta model. b) When decoupling the unified delta model, we use the task trigger $i$ on the unified delta model to reconstruct the delta model $i$.

$\delta^{1:t} = \text{unify}(\{\delta^1, \delta^2, \ldots, \delta^t\})$, where $\delta^{1:t}$ is the unified delta model, and the $j$-th dimension of it is calculated as $\delta_j^{1:t} = \begin{cases} \max_i(\delta_j^i) & \text{if } \sum_{i=1}^t \delta_j^i > 0 \\ \min_i(\delta_j^i) & \text{if } \sum_{i=1}^t \delta_j^i < 0. \end{cases}$ This process means choosing the $j$-th parameter of all 1 to $t$ delta models with the largest absolute value and has the same sign as $\sum_{i=1}^t \delta_j^i$, retaining the largest magnitude and consistent sign information shared across the delta models.

Then the unified delta model is added to the pre-trained VLM to construct the unified model $\theta^{1:t} = \theta^0 + \delta^{1:t}$.

Except unified model, other productions of Unifying Models at session $t$ are $t$ task triggers. For $i = 1, 2, \ldots, t$, each task trigger $i$ will be used on the unified model to reconstruct delta model $i$ in the future, composed of a mask $M^i$ with the same dimension as the delta model, and a rescaling scalar $\lambda^i$. The binary number of $M^i$ at position $j$ indicates whether the delta model $i$ has the same sign as the unified delta model at position $j$, that is $M_j^i = \begin{cases} 1 & \text{if } \delta_j^i \cdot \delta_j^{1:t} > 0 \\ 0 & \text{if } \delta_j^i \cdot \delta_j^{1:t} < 0. \end{cases}$ The rescaler is to preserve the average magnitude of elements in $\delta^i$ and $M^i \odot \delta^{1:t}$, defined as $\lambda^i = \frac{\text{sum}(\text{abs}(\delta^i))}{\text{sum}(\text{abs}(M^i \odot \delta^{1:t}))}$.

The final productions of Unifying Model at session $t$ is a unified model and $t$ task triggers. Then we introduce how to use task triggers to decouple the unified model.

**Decoupling Unified Model.** The process of Decoupling Unified Model is illustrated in 2b. This process is needed in both the beginning of a training session and the inference stage. If there are $t$ seen tasks, $t$ task triggers are applied to the unified delta model to obtain $t$ delta models $\tilde{\delta}^i = \lambda^i \cdot M^i \odot \delta^{1:t}$, which are then added to the pre-trained VLM $\theta^0$ to obtain $t$ task experts $\tilde{\theta}^i = \theta^0 + \tilde{\delta}^i$. At a training session, $\tilde{\delta}^i$ will attend the unifying instead of $\delta^i$. At inference stage, the output logits of all reconstructed task expert $f(\cdot, \tilde{\theta}^i)$ will aggregated to predict the test samples. We will introduce this Aggregating Predictions in the next section.

## 4.2 Semantic-based Aggregating Mechanism at Inference Stage

During inference, we propose a Semantic-based Aggregating Mechanism to predict a sample without task ID or a sample from unseen tasks. Specifically, the unified model is decoupled into task experts by task triggers at the inference stage. For a test sample from a seen task with a known task ID, we choose the corresponding task expert to predict. For a test sample from an unseen task or without task ID, we input the sample into all task experts, and the output logits are added with weights calculated by semantic matching, using memory-stored prototypes computed during the training stage as the next paragraph shows.

**Computing Prototypes.** The process of Computing Prototypes is illustrated in 3. For each category in each task, we save its prototype during training. The prototype of the $k$-th category in the $i$-th task is the mean of the image feature vectors plus the text feature vector for that category, extracted by the pre-trained VLM, that is $P_k^i = f(y, \theta^0) + \frac{1}{|\mathcal{D}_k^t|} \sum_{m=1}^{|\mathcal{D}_k^t|} f(x_m, \theta^0)$, where $\mathcal{D}_k^t$ is the dataset of the $k$-th category in the $i$-th task, $y$ is the text of this category, and $x_m$ is the $m$-th image of $\mathcal{D}_k^t$. Then we will introduce how the aggregating weights are calculated by these prototypes and how they are utilized to aggregate predictions.

**Aggregating Predictions.** The process of Aggregating Predictions is illustrated in Figure 1. For a test image $x$, we use $f(\cdot, \theta^0)$ to extract the image feature. We then calculate the cosine similarity between the test image feature and the learned prototypes of different tasks. For each task, we select the highest similarity score as the weight of this task expert for the test sample. We compare the weights across different task experts and reassign the weights of the K-highest tasks to 1, and the others to 0. The output logits of all task experts are added with these weights to determine the final prediction result. This value $K$ is the only hyperparameter that needs to be determined in our method, and the ablation study of $K$ is in Appendix F, which shows that the performance of our method is very insensitive to the choice of $K$. The inference time of Aggregating Predictions is very close to that of inference by a single model, since the time taken by the model selection phase is almost negligible, and the forward propagation of multiple task experts can be computed in parallel (see Appendix I for a detailed analysis of inference time).

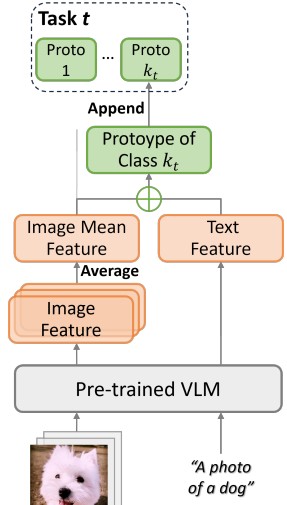

Figure 3: The process of Computing Prototypes: The prototype of each category is the mean of the image feature vectors plus the text feature vector for that category, all extracted by the original pre-trained VLM.

## 5 Experiments

### 5.1 Experiment Setting

We apply ConDU to two fine-tuning scenarios. **ConDU (FT)**: Apply ConDU to full parameter fine-tuning. **ConDU (LoRA)**: Freeze the pre-trained model parameters and only fine-tune the parameters of LoRA modules during the ConDU process. LoRA is one of the most commonly used PEFT modules, and its detailed introduction is in Appendix L. We test our method on three benchmarks, including Multi-domain Task Incremental Learning (MTIL) (Zheng et al., 2023), task-agnostic MTIL, and few-shot MTIL. The MTIL (Zheng et al., 2023) extends task incremental learning to a cross-domain setting, where each task is derived from a distinct domain, including 11 individual tasks. As in (Zheng et al., 2023; Park, 2024; Yu et al., 2024; Xu et al., 2024; Zhang et al., 2024), the tasks are arranged alphabetically, and we further report how ConDU outperforms SOTA methods under another task order in Appendix C. The task-agnostic MTIL is the variant of the MTIL benchmark, where the task ID is unknown during inference for each test sample. The few-shot MTIL variant involves training with only five train samples per category for each task. We follow the existing works (Zheng et al., 2023; Park, 2024; Yu et al., 2024; Li et al., 2024) to use three key metrics for evaluation. The "Transfer" metric evaluates the model's zero-shot transfer performance on subsequent tasks. The "Average" metric evaluates the model's performance on specific task averaged over all sessions, regardless of whether the task has already been encountered in a particular session. The

Table 1: Comparison with SOTA methods on MTIL benchmark in terms of "Transfer", "Average", and "Last" scores (%). We label the best methods on average of all datasets with **bold** styles. The lines with background color represent our methods. The results of more baselines can be found in Appendix B.

| | Method | Aircraft | Caltech101 | CIFAR100 | DTD | EuroSAT | Flowers | Food | MNIST | OxfordPet | Cars | SUN397 | *Average* |
|---|---|---|---|---|---|---|---|---|---|---|---|---|---|
| | Zero-shot | 24.3 | 88.4 | 68.2 | 44.6 | 54.9 | 71.0 | 88.5 | 59.4 | 89.0 | 64.7 | 65.2 | 65.3 |
| | Individual FT | 62.0 | 95.1 | 89.6 | 79.5 | 98.9 | 97.5 | 92.7 | 99.6 | 94.7 | 89.6 | 81.8 | 89.2 |
| Transfer | ZSCL | - | 86.0 | 67.4 | 45.4 | 50.4 | 69.1 | 87.6 | 61.8 | 86.8 | 60.1 | 66.8 | 68.1 |
| | Dual-RAIL | - | 88.4 | 68.2 | 44.6 | 54.9 | 71.0 | 88.5 | 59.6 | 89.0 | 64.7 | 65.2 | 69.4 |
| | DPeCLIP | - | 88.2 | 67.2 | 44.7 | 54.0 | 70.6 | 88.2 | 59.5 | 89.0 | 64.7 | 64.8 | 69.1 |
| | MulKI | - | 87.8 | 69.0 | 46.7 | 51.8 | 71.3 | 88.3 | 64.7 | 89.7 | 63.4 | 68.1 | 70.1 |
| | ConDU (LoRA) | - | 88.1 | 68.9 | 45.7 | 57.0 | 71.3 | 88.8 | 61.2 | 89.3 | 65.1 | 67.8 | 70.3 |
| | ConDU (FT) | - | 88.1 | 68.9 | 46.4 | 57.1 | 71.4 | 88.7 | 65.5 | 89.3 | 65.0 | 67.8 | **70.8** |
| Average | ZSCL | 45.1 | 92.0 | 80.1 | 64.3 | 79.5 | 81.6 | 89.6 | 75.2 | 88.9 | 64.7 | 68.0 | 75.4 |
| | Dual-RAIL | 52.5 | 96.0 | 80.6 | 70.4 | 81.3 | 86.3 | 89.1 | 73.9 | 90.2 | 68.5 | 66.5 | 77.8 |
| | DPeCLIP | 49.9 | 94.9 | 82.4 | 69.4 | 82.2 | 84.3 | 90.0 | 74.0 | 90.4 | 68.3 | 66.3 | 77.5 |
| | MulKI | 52.5 | 93.6 | 79.4 | 67.0 | 79.8 | 83.9 | 89.6 | 77.1 | 91.2 | 67.1 | 69.1 | 77.3 |
| | ConDU (LoRA) | 51.9 | 94.9 | 84.4 | 69.8 | 81.1 | 84.4 | 90.0 | 77.3 | 89.5 | 69.0 | 69.3 | 78.3 |
| | ConDU (FT) | 59.6 | 93.4 | 83.7 | 68.1 | 83.4 | 83.7 | 90.1 | 76.7 | 90.6 | 68.6 | 68.6 | **78.8** |
| Last | ZSCL | 40.6 | 92.2 | 81.3 | 70.5 | 94.8 | 90.5 | 91.9 | 98.7 | 93.9 | 85.3 | 80.2 | 83.6 |
| | Dual-RAIL | 52.5 | 96.8 | 83.3 | 80.1 | 96.4 | 99.0 | 89.9 | 98.8 | 93.5 | 85.5 | 79.2 | 86.8 |
| | DPeCLIP | 49.9 | 95.6 | 85.8 | 78.6 | 98.4 | 95.8 | 92.1 | 99.4 | 94.0 | 84.5 | 81.7 | 86.9 |
| | MulKI | 49.7 | 93.0 | 82.8 | 73.7 | 96.2 | 92.3 | 90.4 | 99.0 | 94.8 | 85.2 | 78.9 | 85.1 |
| | ConDU (LoRA) | 48.9 | 95.2 | 87.8 | 78.5 | 96.3 | 95.2 | 91.7 | 97.6 | 93.0 | 85.3 | 78.8 | 86.2 |
| | ConDU (FT) | 58.6 | 93.7 | 86.6 | 76.1 | 98.2 | 93.4 | 91.9 | 99.6 | 94.8 | 84.9 | 80.5 | **87.1** |

"Last" metric reflects the model's average performance at the end of the continual learning process. In the task-agnostic MTIL setting, we omit the "Transfer" and focus solely on the "Average" and "Last". More implementation details and description of baselines are provided in Appendix A. To explore the generality of the ConDU framework beyond vision-language models, we also conducted additional experiments on single-modality class-incremental learning in Appendix D.

## 5.2 COMPARISON WITH STATE-OF-THE-ART METHODS

**Multi-Domain Task Incremental Learning.** Table 1 presents the detailed comparison results of our proposed ConDU and the baselines on the MTIL benchmark. As seen, our method outperforms all baseline methods across all three metrics. The "Transfer" metric of our method is 70.8% for FT and 70.3% for LoRA, which exceeds the best baseline by 0.7% and surpasses the pre-trained VLM by 5.5%. The "Average" metric of our method is 78.8% for FT and 78.3% for LoRA, which exceeds the best baseline by 1.5% and surpasses the pre-trained VLM by 13.5%. The "Last" metric of our method is 87.1% for FT and 86.2% for LoRA, which exceeds the best baseline by 0.2% and surpasses the pre-trained VLM by 21.9%. These results highlight our approach's effectiveness in mitigating catastrophic forgetting while progressively incorporating new knowledge.

**Task-Agnostic MTIL.** Table 2 presents the detailed comparison results of our proposed ConDU and the baselines on the task-agnostic MTIL benchmark. As seen, our method outperforms all baseline methods across all two metrics. The "Average" metric of our method is 78.1% for FT and 78.0% for LoRA, which exceeds the best baseline by 2% and surpasses pre-trained VLM by 20.3%. The "Last" metric of our method is 86.4% for FT and 85.1% for LoRA, which exceeds the best baseline by 1.8% and surpasses the pre-trained VLM by 28.6%. These results highlight our approach's effectiveness in mitigating catastrophic forgetting while progressively incorporating new knowledge even without a task ID.

**Few-Shot MTIL.** Table 3 presents the comparison results of ConDU and the baselines on the few-shot MTIL benchmark. The "Transfer" metric of our method is 70.0% for FT and 70.3% for LoRA, which exceeds the best baseline by 1.4% and surpasses the pre-trained VLM by 4.7%. The "Average" metric of our method is 72.3% for FT and 72.7% for LoRA, which exceeds the best baseline by 1.3% and surpasses the pre-trained VLM by 7.4%. The "Last" metric of our method is 76.6% for FT and 77.4% for LoRA, which exceeds the best baseline by 1.3% and surpasses the pre-trained VLM by 12.1%. These results highlight our approach's effectiveness in mitigating catastrophic forgetting while progressively incorporating new knowledge even with very few samples.

 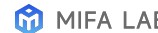

Table 2: Comparison with SOTA methods on task-agnostic MTIL benchmark in terms of "Transfer", "Average", and "Last" scores (%). We label the best methods on average of all datasets with **bold** styles. The lines with background color represent our methods. Individual FT can not be utilized on task-agnostic MTIL, so the Individual FT results here is the prediction with task ID while other methods cannot know the task ID.

| | Method | Aircraft | Caltech101 | CIFAR100 | DTD | EuroSAT | Flowers | Food | MNIST | OxfordPet | Cars | SUN397 | Average |
|---|---|---|---|---|---|---|---|---|---|---|---|---|---|
| | Zero-shot | 24.4 | 63.7 | 41.0 | 39.3 | 53.0 | 70.0 | 88.4 | 39.6 | 88.9 | 64.5 | 63.3 | 57.8 |
| | Individual FT | 62.0 | 95.1 | 89.6 | 79.5 | 98.9 | 97.5 | 92.7 | 99.6 | 94.7 | 89.6 | 81.8 | 89.2 |
| **Average** | Continual FT | 25.5 | 81.5 | 59.1 | 53.2 | 64.7 | 51.8 | 63.2 | 64.3 | 69.7 | 31.8 | 49.7 | 55.9 |
| | ZSCL | 46.3 | 68.3 | 74.3 | 56.3 | 79.1 | 81.4 | 89.5 | 74.0 | 89.0 | 64.4 | 67.5 | 71.8 |
| | MoE | 37.2 | 65.3 | 79.5 | 67.6 | 19.7 | 83.1 | 80.5 | 74.0 | 88.5 | 67.5 | 65.3 | 66.2 |
| | Primal-RAIL | 42.4 | 88.5 | 57.1 | 55.7 | 64.7 | 80.7 | 83.0 | 62.9 | 84.8 | 68.7 | 63.7 | 68.4 |
| | Dual-RAIL | 45.0 | 88.8 | 57.8 | 56.8 | 66.2 | 81.0 | 85.2 | 63.4 | 87.8 | 68.9 | 64.7 | 69.6 |
| | CoLeCLIP | 48.2 | 77.8 | 71.7 | 65.7 | 76.8 | 83.8 | 89.6 | 72.2 | 90.3 | 68.0 | 66.4 | 73.7 |
| | DPeCLIP | 49.9 | 85.3 | 81.5 | 65.3 | 81.6 | 84.3 | 89.9 | 74.0 | 90.4 | 68.3 | 66.2 | 76.1 |
| | ConDU (LoRA) | 51.8 | 94.4 | 84.2 | 68.8 | 80.0 | 84.1 | 90.0 | 77.1 | 88.9 | 68.8 | 69.3 | 78.0 |
| | ConDU (FT) | 59.7 | 90.4 | 83.6 | 67.0 | 81.8 | 83.6 | 90.2 | 75.0 | 90.8 | 68.7 | 68.4 | **78.1** |
| **Last** | Continual FT | 31.0 | 89.3 | 65.8 | 67.3 | 88.9 | 71.1 | 85.6 | 99.6 | 92.9 | 77.3 | 81.1 | 77.3 |
| | ZSCL | 42.5 | 64.4 | 67.2 | 54.8 | 89.7 | 90.4 | 91.7 | 95.8 | 93.4 | 85.2 | 78.3 | 77.6 |
| | MoE | 34.1 | 47.6 | 80.9 | 75.5 | 0.0 | 93.0 | 70.8 | 99.4 | 86.4 | 79.8 | 68.9 | 66.9 |
| | Primal-RAIL | 41.9 | 94.0 | 73.7 | 67.8 | 84.4 | 97.0 | 83.4 | 92.6 | 86.9 | 75.7 | 71.4 | 79.0 |
| | Dual-RAIL | 45.2 | 94.4 | 74.7 | 70.7 | 87.3 | 97.9 | 86.5 | 92.8 | 91.9 | 81.7 | 76.7 | 81.8 |
| | CoLeCLIP | 48.1 | 73.1 | 65.2 | 69.6 | 84.0 | 96.2 | 90.9 | 94.6 | 93.5 | 82.6 | 79.3 | 79.7 |
| | DPeCLIP | 49.9 | 84.2 | 83.2 | 71.1 | 97.0 | 95.8 | 92.0 | 99.4 | 93.9 | 84.5 | 80.2 | 84.6 |
| | ConDU (LoRA) | 48.4 | 94.4 | 87.3 | 77.1 | 94.1 | 94.3 | 90.8 | 96.2 | 90.8 | 84.3 | 78.1 | 85.1 |
| | ConDU (FT) | 58.6 | 90.8 | 86.3 | 74.0 | 96.3 | 93.4 | 91.9 | 99.6 | 94.7 | 84.9 | 80.1 | **86.4** |

Table 3: Comparison with SOTA methods on few-shot MTIL benchmark in terms of "Transfer", "Average", and "Last" scores (%). We label the best methods on average of all datasets with **bold** styles. The lines with background color represent our methods. The results of more baselines can be found in Appendix B.

| | Method | Aircraft | Caltech101 | CIFAR100 | DTD | EuroSAT | Flowers | Food | MNIST | OxfordPet | Cars | SUN397 | Average |
|---|---|---|---|---|---|---|---|---|---|---|---|---|---|
| | Zero-shot | 24.3 | 88.4 | 68.2 | 44.6 | 54.9 | 71.0 | 88.5 | 59.6 | 89.0 | 64.7 | 65.2 | 65.3 |
| | Individual FT | 30.6 | 93.5 | 76.8 | 65.1 | 91.7 | 92.9 | 83.3 | 96.6 | 84.9 | 65.4 | 71.3 | 77.5 |
| **Transfer** | Continual FT | - | 72.8 | 53.0 | 36.4 | 35.4 | 43.3 | 68.4 | 47.4 | 72.6 | 30.0 | 52.7 | 51.2 |
| | WiSE-FT | - | 77.6 | 60.0 | 41.3 | 39.4 | 53.0 | 76.6 | 58.1 | 75.5 | 37.3 | 58.2 | 57.7 |
| | ZSCL | - | 84.0 | 68.1 | 44.8 | 46.8 | 63.6 | 84.9 | 61.4 | 81.4 | 55.5 | 62.2 | 65.3 |
| | MoE | - | 87.9 | 68.2 | 44.1 | 48.1 | 64.7 | 88.8 | 69.0 | 89.1 | 64.5 | 65.1 | 68.9 |
| | ConDU (FT) | - | 88.0 | 69.5 | 45.6 | 54.4 | 71.1 | 88.7 | 62.2 | 88.9 | 64.4 | 66.6 | 70.0 |
| | ConDU (LoRA) | - | 88.1 | 68.5 | 45.6 | 56.4 | 71.2 | 89.0 | 64.0 | 88.8 | 64.9 | 66.4 | **70.3** |
| **Average** | Continual FT | 28.1 | 86.4 | 59.1 | 52.8 | 55.8 | 62.0 | 70.2 | 64.7 | 75.5 | 35.0 | 54.0 | 58.5 |
| | WiSE-FT | 32.0 | 87.7 | 61.0 | 55.8 | 68.1 | 69.3 | 76.8 | 71.5 | 77.6 | 42.0 | 59.3 | 63.7 |
| | ZSCL | 28.2 | 88.6 | 66.5 | 53.5 | 56.3 | 73.4 | 83.1 | 56.4 | 82.4 | 57.5 | 62.9 | 64.4 |
| | MoE | 30.0 | 88.6 | 73.9 | 58.7 | 69.3 | 79.3 | 88.1 | 76.5 | 89.1 | 65.3 | 65.8 | 71.4 |
| | ConDU (FT) | 33.1 | 90.5 | 74.1 | 58.3 | 76.2 | 81.0 | 87.9 | 73.4 | 88.0 | 64.8 | 67.1 | 72.3 |
| | ConDU (LoRA) | 32.4 | 92.1 | 75.4 | 58.8 | 75.1 | 82.9 | 87.3 | 74.0 | 89.3 | 65.1 | 67.0 | **72.7** |
| **Last** | Continual FT | 27.8 | 86.9 | 60.1 | 58.4 | 56.6 | 75.7 | 73.8 | 93.1 | 82.5 | 57.0 | 66.8 | 67.1 |
| | WiSE-FT | 30.8 | 88.9 | 59.6 | 60.3 | 80.9 | 81.7 | 77.1 | 94.9 | 83.2 | 62.8 | 70.0 | 71.9 |
| | ZSCL | 26.8 | 88.5 | 63.7 | 55.7 | 60.2 | 82.1 | 82.6 | 58.6 | 85.9 | 66.7 | 70.4 | 67.4 |
| | MoE | 30.1 | 89.3 | 74.9 | 64.0 | 82.3 | 89.4 | 87.1 | 89.0 | 89.1 | 69.5 | 72.5 | 76.1 |
| | ConDU (FT) | 33.3 | 90.7 | 75.0 | 63.1 | 88.8 | 88.6 | 87.0 | 91.8 | 85.6 | 66.5 | 71.9 | 76.6 |
| | ConDU (LoRA) | 31.8 | 92.4 | 76.7 | 63.4 | 86.8 | 91.8 | 85.6 | 93.9 | 90.3 | 68.1 | 70.9 | 77.4 |

## 5.3 ABLATION STUDY

**PTM vs. Task Expert Features.** We compare prototype–sample similarity computed via (a) shared PTM features and (b) task-specific expert features. In the latter variant, prototypes and test samples are mapped into the feature space of the corresponding task expert for similarity computation. Results show that our PTM-based strategy consistently outperforms the expert-based approach. This indicates that the frozen PTM provides a more unified and reliable representation for cross-task similarity compared to disjoint expert-specific spaces.

**The Effect of Rescalers.** We compared ConDU (FT) with its no-rescaler variant. The variant performs substantially worse, confirming the necessity of rescaling. Importantly, without the rescaler,

the reconstructed task experts will produce features with significantly mismatched magnitudes compared to the original task experts, motivating the inclusion of rescalers.

Table 4: Comparison of "Transfer" and "Average" metrics on MTIL benchmark between PTM-based and Expert-based feature extraction approaches.

| | Method | Aircraft | Caltech101 | CIFAR100 | DTD | EuroSAT | Flowers | Food | MNIST | OxfordPet | Cars | SUN397 | *Average* |
|---|---|---|---|---|---|---|---|---|---|---|---|---|---|
| **Transfer** | Expert-based | - | 88.2 | 68.9 | 40.3 | 49.1 | 69.7 | 86.4 | 62.9 | 84.6 | 59.7 | 67.4 | 67.7 |
| | PTM-based | - | 88.1 | 68.9 | 46.4 | 57.0 | 71.3 | 88.7 | 65.5 | 89.3 | 65.0 | 67.8 | 70.8 |
| **Average** | Expert-based | 59.6 | 93.4 | 83.7 | 67.5 | 81.7 | 82.7 | 89.1 | 77.0 | 88.2 | 65.1 | 68.3 | 77.8 |
| | PTM-based | 59.6 | 93.4 | 83.7 | 68.1 | 83.4 | 83.7 | 90.1 | 76.7 | 90.6 | 68.6 | 68.6 | 78.8 |

Table 5: Comparison of "Transfer", "Average" and "Last" metrics on MTIL benchmark between ConDU (FT) and its no-rescaler variant.

| | Method | Aircraft | Caltech101 | CIFAR100 | DTD | EuroSAT | Flowers | Food | MNIST | OxfordPet | Cars | SUN397 | *Average* |
|---|---|---|---|---|---|---|---|---|---|---|---|---|---|
| **Transfer** | ConDU | - | 88.1 | 68.9 | 46.4 | 57.0 | 71.3 | 88.7 | 65.5 | 89.3 | 65.0 | 67.8 | 70.8 |
| | - w/o rescalers | - | 88.2 | 68.7 | 46.1 | 56.5 | 71.4 | 88.7 | 63.8 | 89.1 | 64.9 | 66.7 | 70.4 |
| **Average** | ConDU | 59.6 | 93.4 | 83.7 | 68.1 | 83.4 | 83.7 | 90.1 | 76.7 | 90.6 | 68.6 | 68.6 | 78.8 |
| | - w/o rescalers | 59.2 | 90.8 | 81.5 | 62.2 | 82.5 | 52.0 | 88.8 | 76.8 | 87.8 | 66.7 | 67.8 | 74.2 |
| **Last** | ConDU | 58.6 | 93.7 | 86.6 | 76.1 | 98.2 | 93.4 | 91.9 | 99.6 | 94.8 | 84.9 | 80.5 | 87.1 |
| | - w/o rescalers | 57.8 | 89.9 | 83.1 | 66.2 | 96.9 | 30.3 | 88.7 | 99.6 | 84.4 | 74.3 | 77.1 | 77.1 |

## 6 DISCUSSION

**Hardware Robustness.** We conducted experiments (ConDU FT) on NVIDIA RTX 4090 and Huawei Ascend 910B to verify the hardware robustness. As shown in Table 6, the discrepancy remains negligible.

**Computational Cost and Storage Analysis.** We evaluate the efficiency of ConDU by comparing learnable parameters with SOTA methods (Figure 4). ConDU (LoRA) requires significantly fewer parameters while achieving superior performance, whereas ConDU (FT) delivers the best performance among full-parameter update methods with a comparable parameter count. Regarding storage, ConDU

Table 6: Performance comparison across different hardware platforms.

| | Transfer | Average | Last |
|---|---|---|---|
| Ascend 910B | 70.2 | 79.0 | 87.0 |
| RTX 4090 | 70.8 | 78.8 | 87.1 |

significantly alleviates the overhead of Individual FT, with efficiency gains scaling as tasks and finetunable parameters increase (Appendix H). Furthermore, ConDU maintains competitive training and inference times (Appendix I): it matches Continual FT in training efficiency, saves 62% time compared to ZSCL (Zheng et al., 2023), and retains inference speeds comparable to a single model.

*t*-SNE Visualization of Feature Space. To compare the change of feature space of task experts from initial fine-tuning to the end of all sessions, we perform *t*-SNE visualization of the features extracted from the training data of Task 1 (AirCraft). Figure 5a illustrates that after session 1, the fine-tuned task expert 1 shows significantly better data discrimination on Task 1. Figure 5b demonstrates that throughout continual learning process by ConDU (FT), task expert 1 undergoes multiple rounds of unifying and decoupling, but its feature space changes very little, almost undetectable by *t*-SNE. This indicates that the task expert reconstructed by ConDU closely matches the representation ability of the model obtained through initial fine-tuning.

**Convergence of Delta Models.** We further theoretically analyzed that in the ConDU process, when the number of sessions approaches infinity, the change of each task expert parameter is monotonically non-increasing. The proof of this theorem is in Appendix G.

**Theorem 1** (Convergence of Delta Models). *Suppose the relative order of rescalers remains invariant throughout the continual learning process. For any session $t \geq 1$, we have $t-1$ delta models*

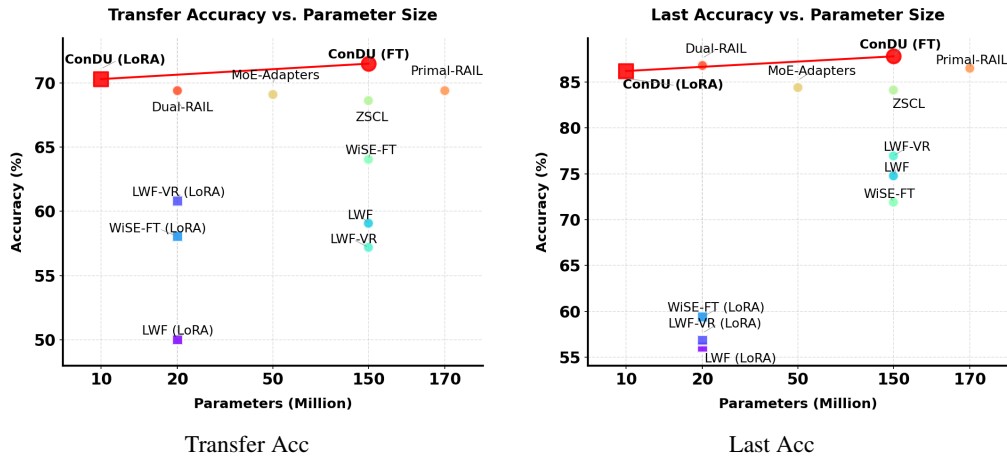

Transfer Acc

Last Acc

Figure 4: Comparison of parameter-accuracy trade-off with SOTA continual learning methods. The vertical axis represents the evaluation metrics of different continual learning methods (namely "Transfer" and "Last" in the two subplots), while the horizontal axis indicates the logarithm of the number of learnable parameters during the training process for each method.

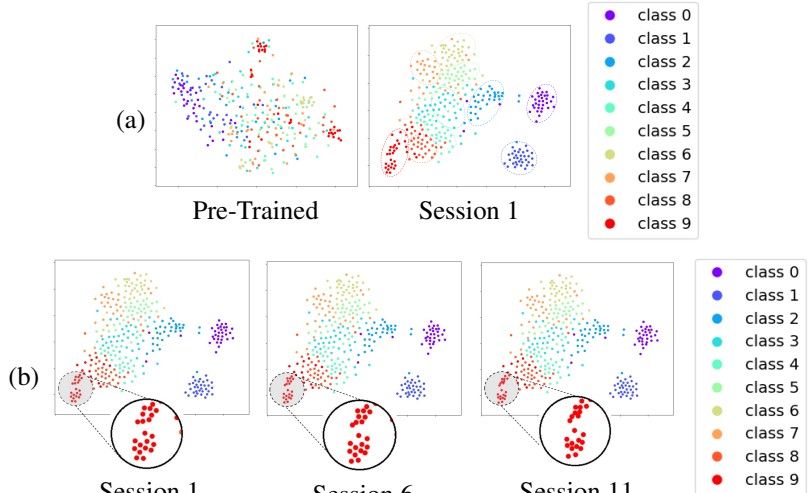

Figure 5: We perform $t$-SNE visualization of the features extracted from the training data of Task 1 (AirCraft). We use four models in total, including the pre-trained VLM and the task expert 1 fine-tuned at the end of sessions 1 and reconstructed by ConDU (FT) at the end of sessions 6 and 11, to extract features of 10 randomly sampled data categories in Task 1. (a) We perform the t-SNE visualization on pre-trained model and task expert 1 independently to provide fair comparison. (b) We concatenate the features extracted from task expert 1 after session 1, 6, and 11 for t-SNE visualization. The enlarged area is to show the slight changes of features.

$\delta^1(t), \delta^2(t), \ldots, \delta^{t-1}(t)$ *decoupled from a bounded unified delta model, along with the latest delta model* $\delta^t(t)$. *If all delta models are independently and identically distributed, and the parameter signs of all delta models are identical for each dimension, then for each* $i \in \{1, 2, \ldots, t-1\}$, *the following property holds.*

$$\lim_{t \to +\infty} \mathbb{E}\left[\|\delta^i(t+1) - \delta^i(t)\|_1 - \|\delta^i(t) - \delta^i(t-1)\|_1\right] \leq 0.$$

**More Discussion.** We comprehensively compared ConDU (FT) and ConDU (LoRA) in Appendix E to show distinct advantages of full-finetuning and PEFT under different conditions. We also comprehensively compared ConDU and Individual FT in Appendix J to show the effectiveness of decoupling-unifying mechanism. The accuracy of each tasks during each session is shown in Appendix K to demonstrate the reconstruction of our method for task experts.

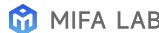

## ACKNOWLEDGMENTS

This project is supported by the National Natural Science Foundation of China (No. 62406192), Shanghai Municipal Special Program for Basic Research on General AI Foundation Models (Grant No. 2025SHZDZX025G03), Opening Project of the State Key Laboratory of General Artificial Intelligence (No. SKLAGI2024OP12), the Tencent WeChat Rhino-Bird Focused Research Program, Kuaishou Technology, and the SJTU Kunpeng & Ascend Center of Excellence.

## REPRODUCIBILITY STATEMENT

Our code is available at https://github.com/zhangzicong518/ConDU. Detailed explanations of the experimental setup can be found in Section 5.1 and Appendix A, where we provide the values of all hyperparameters used during training. Consulting these sections may help in reproducing the results and better understanding our released code.

## ETHICS STATEMENT

This work adheres to the ICLR Code of Ethics. Our research focuses on algorithmic and methodological contributions in continual learning and does not involve human subjects, sensitive personal data, or information that raises direct privacy or security concerns. The datasets used in our experiments (e.g., MTIL, TinyImageNet) are widely adopted public benchmarks released under permissive licenses, and we follow standard usage practices without modification that could introduce ethical risks. The proposed methods are intended for advancing machine learning research and have no foreseeable harmful applications. We are not aware of any conflicts of interest, and the study complies with established principles of fairness, transparency, and research integrity.

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

# Appendix

## A  DETAILED EXPERIMENT SETTING

**Dataset.**  We test our method on three benchmarks, including Multi-domain Task Incremental Learning (MTIL) (Zheng et al., 2023), task-agnostic MTIL, and few-shot MTIL.

The MTIL (Zheng et al., 2023) extends task incremental learning to a cross-domain setting, where each task is derived from a distinct domain. The MTIL framework comprises 11 individual tasks, each associated with a separate dataset, collectively representing a total of 1201 classes. In alignment with previous works, we adopt the following datasets: Aircraft (Maji et al., 2013), Caltech101 (Fei-Fei et al., 2004), CIFAR100 (Krizhevsky et al., 2009), DTD (Cimpoi et al., 2014), EuroSAT (Helber et al., 2019), Flowers (Nilsback & Zisserman, 2008), Food (Bossard et al., 2014), MNIST (Deng, 2012), OxfordPet (Parkhi et al., 2011), StanfordCars (Krause et al., 2013), and SUN397 (Xiao et al., 2010). The task-agnostic MTIL is the variant of the MTIL benchmark, where the task ID is unknown during inference for each test sample. The few-shot MTIL variant involves training with only five train samples per category for each task.

**Protocol.**  In our experiments, all evaluation protocols follow the existing works (Zheng et al., 2023; Park, 2024; Yu et al., 2024; Li et al., 2024) for fair comparison. We utilize a pre-trained CLIP model with a ViT-B/16 (Dosovitskiy et al., 2020) image encoder. We perform 1000 iterations of training for each task in both MTIL and task-agnostic MTIL. For few-shot MTIL, we train each task for 500 iterations. We use AdamW (Loshchilov & Hutter, 2017) as the optimizer and set the batch size to 32 across all experiments.

**Metric.**  For evaluating the MTIL, task-agnostic MTIL, and few-shot MTIL, we follow the existing works (Zheng et al., 2023; Park, 2024; Yu et al., 2024; Li et al., 2024) to use three key metrics: "Average", "Last", and "Transfer". The "Average" metric computes the average accuracy across all seen tasks. The "Transfer" metric evaluates the model's zero-shot transfer performance on subsequent tasks. The "Last" metric reflects the model's average performance at the end of the continual learning process. In the task-agnostic MTIL setting, we omit the "Transfer" metric and focus solely on the "Average" and "Last" metrics.

**Baseline.**  We compare our method with several state-of-the-art (SOTA) approaches, including:

(1) Zero-shot, (2) Individual FT, (3) Continual FT, (4) LwF (Li & Hoiem, 2017), (5) iCaRL (Rebuffi et al., 2017), (6) LwF-VR (Ding et al., 2022), (7) WiSE-FT (Wortsman et al., 2022) (8) ZSCL (Zheng et al., 2023), (9) MoE (Park, 2024), (10) MA (Yu et al., 2024), (11) Primal-RAIL (Xu et al., 2024), (12) Dual-RAIL (Xu et al., 2024), (13) CoLeCLIP (Li et al., 2024), (14) DPeCLIP (Lu et al., 2024), (15) MulKI (Zhang et al., 2024).

The results of baselines (2)–(8) and (15) in this paper are FFT-based. The results of baselines (9)–(14) in this paper are PEFT-based.

Zero-shot denotes directly using the pre-trained VLM for prediction on each task without additional fine-tuning. The results of Individual FT represent the performance of using a fully fine-tuned model, trained independently on each task based on the pre-trained VLM, for prediction. Continual FT refers to incrementally fine-tuning the VLM on new tasks without employing any forgetting mitigation strategies.

## B  MORE RESULTS OF COMPARISON WITH STATE-OF-THE-ART METHODS

**MTIL.**  Table 7 presents the full version of Table 1.

**Few-shot MTIL.**  Table 8 presents the full version of Table 3.

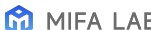 MIFA LAB

Table 7: Comparison with SOTA methods on MTIL benchmark in terms of "Transfer", "Average.", and "Last" scores (%). We label the best methods on average of all datasets with **bold** styles. The lines with background color represent our methods.

| | Method | Aircraft | Caltech101 | CIFAR100 | DTD | EuroSAT | Flowers | Food | MNIST | OxfordPet | Cars | SUN397 | Average |
|---|---|---|---|---|---|---|---|---|---|---|---|---|---|
| | Zero-shot | 24.3 | 88.4 | 68.2 | 44.6 | 54.9 | 71.0 | 88.5 | 59.4 | 89.0 | 64.7 | 65.2 | 65.3 |
| | Individual FT | 62.0 | 95.1 | 89.6 | 79.5 | 98.9 | 97.5 | 92.7 | 99.6 | 94.7 | 89.6 | 81.8 | 89.2 |
| Transfer | Continual FT | - | 67.1 | 46.0 | 32.1 | 35.6 | 35.0 | 57.7 | 44.1 | 60.8 | 20.5 | 46.6 | 44.6 |
| | LwF | - | 74.5 | 56.9 | 39.1 | 51.1 | 52.6 | 72.8 | 60.6 | 75.1 | 30.3 | 55.9 | 58.9 |
| | iCaRL | - | 56.6 | 44.6 | 32.7 | 39.3 | 46.6 | 68.0 | 46.0 | 77.4 | 31.9 | 60.5 | 50.4 |
| | LwF-VR | - | 77.1 | 61.0 | 40.5 | 45.3 | 54.4 | 74.6 | 47.9 | 76.7 | 36.3 | 58.6 | 57.2 |
| | WiSE-FT | - | 73.5 | 55.6 | 35.6 | 41.5 | 47.0 | 68.3 | 53.9 | 69.3 | 26.8 | 51.9 | 52.3 |
| | ZSCL | - | 86.0 | 67.4 | 45.4 | 50.4 | 69.1 | 87.6 | 61.8 | 86.8 | 60.1 | 66.8 | 68.1 |
| | MoE | - | 88.2 | 66.9 | 44.7 | 54.1 | 70.6 | 88.4 | 59.5 | 89.0 | 64.7 | 65.0 | 69.1 |
| | MA | - | 87.9 | 68.2 | 44.4 | 49.9 | 70.7 | 88.7 | 59.7 | 89.1 | 64.5 | 65.5 | 68.9 |
| | Primal-RAIL | - | 88.4 | 68.2 | 44.6 | 54.9 | 71.0 | 88.5 | 59.6 | 89.0 | 64.7 | 65.2 | 69.4 |
| | Dual-RAIL | - | 88.4 | 68.2 | 44.6 | 54.9 | 71.0 | 88.5 | 59.6 | 89.0 | 64.7 | 65.2 | 69.4 |
| | CoLeCLIP | - | 88.2 | 65.1 | 44.7 | 54.1 | 68.8 | 88.5 | 59.5 | 89.0 | 64.7 | 65.1 | 68.8 |
| | DPeCLIP | - | 88.2 | 67.2 | 44.7 | 54.0 | 70.6 | 88.2 | 59.5 | 89.0 | 64.7 | 64.8 | 69.1 |
| | MulKI | - | 87.8 | 69.0 | 46.7 | 51.8 | 71.3 | 88.3 | 64.7 | 89.7 | 63.4 | 68.1 | 70.1 |
| | ConDU (LoRA) | - | 88.1 | 68.9 | 45.7 | 57.0 | 71.3 | 88.8 | 61.2 | 89.3 | 65.1 | 67.8 | 70.3 |
| | ConDU (FT) | - | 88.1 | 68.9 | 46.4 | 57.1 | 71.4 | 88.7 | 65.5 | 89.3 | 65.0 | 67.8 | **70.8** |
| Average | Continual FT | 25.5 | 81.5 | 59.1 | 53.2 | 64.7 | 51.8 | 63.2 | 64.3 | 69.7 | 31.8 | 49.7 | 55.9 |
| | LwF | 36.3 | 86.9 | 72.0 | 59.0 | 73.7 | 60.0 | 73.6 | 74.8 | 80.0 | 37.3 | 58.1 | 64.7 |
| | iCaRL | 35.5 | 89.2 | 72.2 | 60.6 | 68.8 | 70.0 | 78.2 | 62.3 | 81.8 | 41.2 | 62.5 | 65.7 |
| | LwF-VR | 29.6 | 87.7 | 74.4 | 59.5 | 72.4 | 63.6 | 77.0 | 66.7 | 81.2 | 43.7 | 60.7 | 65.1 |
| | WiSE-FT | 26.7 | 86.5 | 64.3 | 57.1 | 65.7 | 58.7 | 71.1 | 70.5 | 75.8 | 54.6 | 60.7 | 60.7 |
| | ZSCL | 45.1 | 92.0 | 80.1 | 64.3 | 79.5 | 81.6 | 89.6 | 75.2 | 88.9 | 64.7 | 68.0 | 75.4 |
| | MoE | 37.4 | 93.9 | 80.5 | 68.3 | 81.9 | 84.1 | 90.0 | 74.0 | 90.6 | 67.7 | 66.4 | 75.9 |
| | MA | 50.2 | 91.9 | 83.1 | 69.4 | 78.9 | 84.0 | 89.1 | 73.7 | 89.3 | 67.7 | 66.9 | 76.7 |
| | Primal-RAIL | 51.9 | 95.8 | 80.1 | 70.3 | 81.1 | 86.1 | 89.0 | 73.9 | 90.2 | 68.4 | 66.4 | 77.6 |
| | Dual-RAIL | 52.5 | 96.0 | 80.6 | 70.4 | 81.3 | 86.3 | 89.1 | 73.9 | 90.2 | 68.5 | 66.5 | 77.8 |
| | CoLeCLIP | 48.7 | 94.3 | 76.6 | 69.2 | 79.0 | 83.8 | 89.7 | 73.3 | 90.5 | 68.0 | 66.5 | 76.3 |
| | DPeCLIP | 49.9 | 94.9 | 82.4 | 69.4 | 82.2 | 84.3 | 90.0 | 74.0 | 90.4 | 68.3 | 66.3 | 77.5 |
| | MulKI | 52.5 | 93.6 | 79.4 | 67.0 | 79.8 | 83.9 | 89.6 | 77.1 | 91.2 | 67.1 | 69.1 | 77.3 |
| | ConDU (LoRA) | 51.9 | 94.9 | 84.4 | 69.8 | 81.1 | 84.4 | 90.0 | 77.3 | 89.5 | 69.0 | 69.3 | 78.3 |
| | ConDU (FT) | 59.6 | 93.4 | 83.7 | 68.1 | 83.4 | 83.7 | 90.1 | 76.7 | 90.6 | 68.6 | 68.6 | **78.8** |
| Last | Continual FT | 31.0 | 89.3 | 65.8 | 67.3 | 88.9 | 71.1 | 85.6 | 99.6 | 92.9 | 77.3 | 81.1 | 77.3 |
| | LwF | 26.3 | 87.5 | 71.9 | 66.6 | 79.9 | 66.9 | 83.8 | 99.6 | 92.1 | 66.1 | 80.4 | 74.6 |
| | iCaRL | 35.8 | 93.0 | 77.0 | 70.2 | 83.3 | 88.5 | 90.4 | 86.7 | 93.2 | 81.2 | 81.9 | 80.1 |
| | LwF-VR | 20.5 | 89.8 | 72.3 | 67.6 | 85.5 | 73.8 | 85.7 | 99.6 | 93.1 | 73.3 | 80.9 | 76.6 |
| | WiSE-FT | 27.2 | 90.8 | 68.0 | 68.9 | 86.9 | 74.0 | 87.6 | 99.6 | 92.6 | 77.8 | 81.3 | 77.7 |
| | ZSCL | 40.6 | 92.2 | 81.3 | 70.5 | 94.8 | 90.5 | 91.9 | 98.7 | 93.9 | 85.3 | 80.2 | 83.6 |
| | MoE | 34.6 | 94.7 | 82.7 | 76.9 | 97.7 | 94.8 | 91.9 | 99.4 | 94.7 | 80.9 | 80.5 | 84.4 |
| | MA | 49.8 | 92.2 | 86.1 | 78.1 | 95.7 | 94.3 | 89.5 | 98.1 | 89.9 | 81.6 | 80.0 | 85.0 |
| | Primal-RAIL | 51.9 | 96.5 | 82.8 | 80.0 | 96.0 | 98.7 | 89.7 | 98.8 | 93.3 | 84.8 | 78.7 | 86.5 |
| | Dual-RAIL | 52.5 | 96.8 | 83.3 | 80.1 | 96.4 | 99.0 | 89.9 | 98.8 | 93.5 | 85.5 | 79.2 | 86.8 |
| | CoLeCLIP | 48.7 | 94.9 | 78.8 | 78.4 | 88.9 | 96.3 | 91.1 | 97.6 | 94.4 | 82.7 | 80.2 | 84.7 |
| | DPeCLIP | 49.9 | 95.6 | 85.8 | 78.6 | 98.4 | 95.8 | 92.1 | 99.4 | 94.0 | 84.5 | 81.7 | 86.9 |
| | MulKI | 49.7 | 93.0 | 82.8 | 73.7 | 96.2 | 92.3 | 90.4 | 99.0 | 94.8 | 85.2 | 78.9 | 85.1 |
| | ConDU (LoRA) | 48.9 | 95.2 | 87.8 | 78.5 | 96.3 | 95.2 | 91.7 | 97.6 | 93.0 | 85.3 | 78.8 | 86.2 |
| | ConDU (FT) | 58.6 | 93.7 | 86.6 | 76.1 | 98.2 | 93.4 | 91.9 | 99.6 | 94.8 | 84.9 | 80.5 | **87.1** |

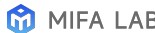

Table 8: Comparison with SOTA methods on few-shot MTIL benchmark in terms of "Transfer", "Average.", and "Last" scores (%). We label the best methods on average of all datasets with **bold** styles. The lines with background color represent our methods.

| | Method | Aircraft | Caltech101 | CIFAR100 | DTD | EuroSAT | Flowers | Food | MNIST | OxfordPet | Cars | SUN397 | *Average* |
|---|---|---|---|---|---|---|---|---|---|---|---|---|---|
| | Zero-shot | 24.3 | 88.4 | 68.2 | 44.6 | 54.9 | 71.0 | 88.5 | 59.6 | 89.0 | 64.7 | 65.2 | 65.3 |
| | Individual FT | 30.6 | 93.5 | 76.8 | 65.1 | 91.7 | 92.9 | 83.3 | 96.6 | 84.9 | 65.4 | 71.3 | 77.5 |
| **Transfer** | Continual FT | - | 72.8 | 53.0 | 36.4 | 35.4 | 43.3 | 68.4 | 47.4 | 72.6 | 30.0 | 52.7 | 51.2 |
| | LwF | - | 72.1 | 49.2 | 35.9 | 44.5 | 41.1 | 66.6 | 50.5 | 69.0 | 19.0 | 51.7 | 50.0 |
| | LwF-VR | - | 82.2 | 62.5 | 40.1 | 40.1 | 56.3 | 80.0 | 60.9 | 77.6 | 40.5 | 60.8 | 60.1 |
| | WiSE-FT | - | 77.6 | 60.0 | 41.3 | 39.4 | 53.0 | 76.6 | 58.1 | 75.5 | 37.3 | 58.2 | 57.7 |
| | ZSCL | - | 84.0 | 68.1 | 44.8 | 46.8 | 63.6 | 84.9 | 61.4 | 81.4 | 55.5 | 62.2 | 65.3 |
| | MoE | - | 87.9 | 68.2 | 44.1 | 48.1 | 64.7 | 88.8 | 69.0 | 89.1 | 64.5 | 65.1 | 68.9 |
| | Primal-RAIL | - | 88.4 | 68.2 | 44.6 | 54.9 | 71.0 | 88.5 | 59.6 | 89.0 | 64.7 | 65.2 | 69.4 |
| | Dual-RAIL | - | 88.4 | 68.2 | 44.6 | 54.9 | 71.0 | 88.5 | 59.6 | 89.0 | 64.7 | 65.2 | 69.4 |
| | ConDU (FT) | - | 88.0 | 69.5 | 45.6 | 54.4 | 71.1 | 88.7 | 62.2 | 88.9 | 64.4 | 66.6 | 70.0 |
| | ConDU (LoRA) | - | 88.1 | 68.5 | 45.6 | 56.4 | 71.2 | 89.0 | 64.0 | 88.8 | 64.9 | 66.4 | **70.3** |
| **Average** | Continual FT | 28.1 | 86.4 | 59.1 | 52.8 | 55.8 | 62.0 | 70.2 | 64.7 | 75.5 | 35.0 | 54.0 | 58.5 |
| | LwF | 23.5 | 77.4 | 43.5 | 41.7 | 43.5 | 52.2 | 54.6 | 63.4 | 68.0 | 21.3 | 52.6 | 49.2 |
| | LwF-VR | 24.9 | 89.1 | 64.2 | 53.4 | 54.3 | 70.8 | 79.2 | 66.5 | 79.2 | 44.1 | 61.6 | 62.5 |
| | WiSE-FT | 32.0 | 87.7 | 61.0 | 55.8 | 68.1 | 69.3 | 76.8 | 71.5 | 77.6 | 42.0 | 59.3 | 63.7 |
| | ZSCL | 28.2 | 88.6 | 66.5 | 53.5 | 56.3 | 73.4 | 83.1 | 56.4 | 82.4 | 57.5 | 62.9 | 64.4 |
| | MoE | 30.0 | 89.6 | 73.9 | 58.7 | 69.3 | 79.3 | 88.1 | 76.5 | 89.1 | 65.3 | 65.8 | 71.4 |
| | Primal-RAIL | 32.9 | 94.5 | 69.9 | 58.1 | 71.8 | 84.4 | 88.5 | 70.4 | 89.0 | 66.1 | 65.7 | 71.9 |
| | Dual-RAIL | 36.0 | 94.2 | 70.9 | 58.8 | 70.6 | 84.3 | 88.5 | 70.3 | 89.7 | 66.5 | 65.8 | 72.3 |
| | ConDU (FT) | 33.1 | 90.5 | 74.1 | 58.3 | 76.2 | 81.0 | 87.9 | 73.4 | 88.0 | 64.8 | 67.1 | 72.3 |
| | ConDU (LoRA) | 32.4 | 92.1 | 75.4 | 58.8 | 75.1 | 82.9 | 87.3 | 74.0 | 89.3 | 65.1 | 67.0 | **72.7** |
| **Last** | Continual FT | 27.8 | 86.9 | 60.1 | 58.4 | 56.6 | 75.7 | 73.8 | 93.1 | 82.5 | 57.0 | 66.8 | 67.1 |
| | LwF | 22.1 | 58.2 | 17.9 | 32.1 | 28.1 | 66.7 | 46.0 | 84.3 | 64.1 | 31.5 | 60.1 | 46.5 |
| | LwF-VR | 22.9 | 89.9 | 59.3 | 57.1 | 57.6 | 79.2 | 78.3 | 77.7 | 83.6 | 60.1 | 69.8 | 66.9 |
| | WiSE-FT | 30.8 | 88.9 | 59.6 | 60.3 | 80.9 | 81.7 | 77.1 | 94.9 | 83.2 | 62.8 | 70.0 | 71.9 |
| | ZSCL | 26.8 | 88.5 | 63.7 | 55.7 | 60.2 | 82.1 | 82.6 | 85.9 | 85.9 | 66.7 | 70.4 | 67.4 |
| | MoE | 30.1 | 89.3 | 74.9 | 64.0 | 82.3 | 89.4 | 87.1 | 89.0 | 89.1 | 69.5 | 72.5 | 76.1 |
| | Primal-RAIL | 32.9 | 95.1 | 70.3 | 63.2 | 81.5 | 95.6 | 88.5 | 89.7 | 89.0 | 72.5 | 71.0 | 77.2 |
| | Dual-RAIL | 36.0 | 94.8 | 71.5 | 64.1 | 79.5 | 95.3 | 88.5 | 89.4 | 91.5 | 74.6 | 71.3 | **77.9** |
| | ConDU (FT) | 33.3 | 90.7 | 75.0 | 63.1 | 88.8 | 88.6 | 87.0 | 91.8 | 85.6 | 66.5 | 71.9 | 76.6 |
| | ConDU (LoRA) | 31.8 | 92.4 | 76.7 | 63.4 | 86.8 | 91.8 | 85.6 | 93.9 | 90.3 | 68.1 | 70.9 | 77.4 |

Table 9: Performance (%) Comparison of of state-of-the-art CL methods on MTIL benchmark in Order II

| Method | Transfer | Δ | Average | Δ | Last | Δ |
|---|---|---|---|---|---|---|
| CLIP Zero-shot (Radford et al., 2021) | 65.4 | 0.0 | 65.3 | 0.0 | 65.3 | 0.0 |
| Continual FT | 46.6 | -18.8 | 56.2 | -9.1 | 67.4 | 2.1 |
| LwF (Li & Hoiem, 2017) | 53.2 | -12.2 | 62.2 | -3.1 | 71.9 | 6.6 |
| iCaRL (Rebuffi et al., 2017) | 50.9 | -14.5 | 56.9 | -8.4 | 71.6 | 6.3 |
| LwF-VR (Ding et al., 2022) | 53.1 | -12.3 | 60.6 | -4.7 | 68.3 | 3.0 |
| WiSE-FT (Wortsman et al., 2022) | 51.0 | -14.4 | 61.5 | -3.8 | 72.2 | 6.9 |
| ZSCL (Zheng et al., 2023) | 64.2 | -1.2 | 74.5 | 9.2 | 83.4 | 18.1 |
| MoE (Park, 2024) | 64.3 | -1.1 | 74.7 | 9.4 | 84.1 | 18.8 |
| MulKI (Zhang et al., 2024) | 65.6 | 0.2 | 75.0 | 9.7 | 84.2 | 18.9 |
| ConDU (FT) | 66.5 | 1.1 | 75.9 | 10.6 | 85.6 | 20.3 |

## C  OTHER ORDER OF MTIL

To further validate the effectiveness of our approach, we refer to existing works (Zheng et al., 2023; Zhang et al., 2024) and present the MTIL experimental results based on another task order in Table 9, with all other experimental settings unchanged. Order II is StanfordCars, Food, MNIST, OxfordPet, Flowers, SUN397, Aircraft, Caltech101, DTD, EuroSAT, CIFAR100. Only a few baselines have reported results for this task order, and we present the results of these baselines in the table. Even with a different task order, we still achieve performance beyond the state-of-the-art.

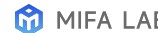

Table 10: Comparison of different methods under 10-step and 20-step settings of TinyImageNet class-Incremental learning.

| Method | 10 Steps | 20 Steps |
|---|---|---|
| ZSCL | 71.62 | 68.30 |
| LwF | 44.00 | 42.26 |
| LwF-VR | 67.05 | 63.89 |
| iCaRL | 65.97 | 64.48 |
| Continual FT | 41.54 | 44.55 |
| CLIP | 65.59 | 65.30 |
| ConDU (FT) | **71.74** | **71.49** |
| ConDU (LoRA) | 68.80 | 68.28 |

## D  EXPERIMENTAL RESULTS ON CLASS-INCREMENTAL LEARNING

To explore the generality of the ConDU framework beyond vision-language models, we conducted additional experiments on a single-modality dataset. Specifically, we evaluated ConDU on TinyImageNet under class-incremental settings with 10, and 20 tasks. The results show that ConDU consistently outperforms competitive baselines, indicating that it is also effective for class-incremental learning in single-modality scenarios.

In the first session, we train the model on 100 categories. Then we split the remaining 100 categories of TinyImageNet dataset into 10 tasks (each with 10 categories) and 20 tasks (each with 5 categories). Following the standard class-incremental learning setup, we train on all tasks sequentially and report the average inference accuracy across all tasks after all sessions (task ID agnostic). The results are in Table 10.

## E  COMPARISON BETWEEN CONDU (FT) AND CONDU (LORA)

**Comparison between the ConDU (FT) and ConDU (LoRA).**   When hardware conditions are limited, we recommend using ConDU (LoRA). When hardware conditions can support full fine-tuning, ConDU (FT) and ConDU (LoRA) has have distinct advantages under different conditions.

- Table 1 (Standard MTIL): For the "Transfer" metric, ConDU (FT) outperforms ConDU (LoRA) on 3 tasks, underperforms on 2, and ties on the rest. Average accuracy favors ConDU (FT). For the "Average" metric, ConDU (FT) outperforms ConDU (LoRA) on 4 tasks, and underperforms on 7. However, again, the average accuracy is higher for ConDU (FT). For the "Last" metric, ConDU (FT) outperforms on 6 tasks and underperforms on 5, with higher average accuracy. Overall, ConDU (FT) achieves higher mean accuracy across all three metrics in standard MTIL, and shows better per-task performance in two metrics out of the three.

- Table 2 (Task-Agnostic MTIL): For the "Average" metric, ConDU (FT) outperforms ConDU (LoRA) on 4 tasks and underperforms on 7, but still achieves a higher overall average. For the "Last" metric, FT outperforms on 7 tasks and underperforms on 4, again with a higher overall average. Overall, ConDU (FT) achieves higher mean accuracy across all two metrics in task-agnostic MTIL, and shows better per-task performance in one metric out of the two.

- Table 3 (Few-shot MTIL): ConDU (LoRA) is better when there is less fine-tuning data for a single task as shown in Table 3), because too little data will lead to overfitting during full fine-tuning.

Then, we also conducted experiments on the TinyImageNet class-incremental benchmark, using both 10-task and 20-task splits. The results in Table 10 show that ConDU (FT) significantly outperforms ConDU (LoRA) under this setting.

In summary, full fine-tuning and PEFT each have distinct advantages under different conditions in continual learning.

Table 11: Comparison of "Transfer" metric on MTIL with different choice of $K$ by ConDU (FT).

| | Aircraft | Caltech101 | CIFAR100 | DTD | EuroSAT | Flowers | Food | MNIST | OxfordPet | Cars | SUN397 | *Average* |
|---|---|---|---|---|---|---|---|---|---|---|---|---|
| $K = 1$ | 88.4 | 68.2 | 44.7 | 55.3 | 71.0 | 88.5 | 59.5 | 89.0 | 64.7 | 65.4 | | 69.5 |
| $K = 2$ | 88.1 | 68.6 | 45.7 | 57.8 | 71.1 | 88.7 | 63.5 | 89.1 | 64.8 | 66.5 | | 70.4 |
| $K = 3$ | 88.1 | 68.9 | 46.2 | 57.4 | 71.4 | 88.7 | 64.0 | 89.4 | 64.9 | 67.2 | | 70.6 |
| $K = 4$ | 88.1 | 68.9 | 46.4 | 56.4 | 71.6 | 88.7 | 64.4 | 89.3 | 64.8 | 67.7 | | 70.6 |
| $K = 5$ | 88.1 | 68.9 | 46.4 | 57.0 | 71.3 | 88.7 | 65.5 | 89.3 | 65.0 | 67.8 | | 70.8 |
| $K = 6$ | 88.1 | 68.9 | 46.4 | 57.0 | 71.5 | 88.5 | 64.0 | 88.6 | 65.0 | 67.9 | | 70.6 |
| $K = 7$ | 88.1 | 68.9 | 46.4 | 57.0 | 71.5 | 88.1 | 64.5 | 88.2 | 64.5 | 67.9 | | 70.5 |
| $K = 8$ | 88.1 | 68.9 | 46.4 | 57.0 | 71.5 | 88.1 | 62.7 | 87.8 | 64.5 | 67.7 | | 70.3 |
| $K = 9$ | 88.1 | 68.9 | 46.4 | 57.0 | 71.5 | 88.1 | 62.7 | 87.6 | 64.4 | 67.7 | | 70.2 |
| $K = 10$ | 88.1 | 68.9 | 46.4 | 57.0 | 71.5 | 88.1 | 62.7 | 87.6 | 64.3 | 67.7 | | 70.2 |
| $K = 11$ | 88.1 | 68.9 | 46.4 | 57.0 | 71.5 | 88.1 | 62.7 | 87.6 | 64.3 | 67.7 | | 70.2 |
| Unified | 66.7 | 44.6 | 29.3 | 43.6 | 47.7 | 60.1 | 48.7 | 53.5 | 28.0 | 49.1 | | 47.1 |

## F  THE CHOICE OF $K$ OF AGGREGATING

Table 11 presents the inference performance of the "Transfer" metric of ConDU (FT) on the MTIL benchmark with different values of $K$. The last row "Unified" denotes directly using unified model to predict unseen tasks. From the table, we can observe that when $K$ is set to 1, the Avg performance is the lowest at 69.5%. As $K$ increases, the Avg performance gradually improves, reaching the best result of 70.8% when $K$ is 5. However, after this point, the Avg performance begins to decrease as $K$ increases further. Overall, the performance of the aggregating mechanism is relatively insensitive to the choice of $K$. The results in the last row indicate that directly using the unified model for predicting unseen tasks yields significantly worse performance compared to our proposed aggregating prediction approach. Within our framework, the unified model can be viewed as a compressed storage mechanism designed to avoid storing all task-specific models individually. Its primary purpose is to interact with task triggers to reconstruct all task-specific models, rather than serving as a standalone model for inference. This is because the objective of model fusion was never intended to make the unified model itself directly applicable for inference.

## G  PROOF OF THEOREM 1

### G.1  DEFINITIONS

One iteration of unifying and decoupling is the same as in Section 4, but some symbols need to be slightly changed. Therefore, we redefine the unifying and decoupling process here. At session $t$, if we have $n$ delta models $\delta^1(t), \delta^2(t), \ldots, \delta^n(t)$, the $j$-th position of unified delta model $\delta(t)$ (which is denoted as $\delta^{1:n}(t)$ in Section 4 and is simplified here) is calculated as

$$\delta_j(t) = \begin{cases} \max_i(\delta_j^i(t)) & \text{if } \sum_{i=1}^n \delta_j^i(t) > 0 \\ \min_i(\delta_j^i(t)) & \text{if } \sum_{i=1}^n \delta_j^i(t) < 0. \end{cases}$$

The $j$-th position of mask $M^i(t)$ is calculated as $M_j^i(t) = \begin{cases} 1 & \text{if } \delta_j^i(t) \cdot \delta_j(t) > 0 \\ 0 & \text{if } \delta_j^i(t) \cdot \delta_j(t) < 0. \end{cases}$

The rescaler $\lambda^i(t)$ is calculated as $\lambda^i(t) = \frac{\text{sum}(\text{abs}(\delta^i(t)))}{\text{sum}(\text{abs}(M^i(t) \odot \delta(t)))}$. The rescaler $\lambda^i(t)$ ensures the $\|\delta^i(t)\|_1$ remains unchanged while $t$ increases.

The reconstructed delta model $i$ is calculated as $\delta^i(t+1) = \lambda^i(t) \cdot M^i(t) \odot \delta(t)$.

In the proof below, there are two settings, and we will mark in each lemma and theorem whether it holds in setting 1 or 2.

**Setting 1**: This setting is the same as the normal continual learning process in the main paper. After the decoupling phase of each session $t$, we have $t - 1$ delta models $\delta^1(t), \delta^2(t), \ldots, \delta^{t-1}(t)$, and then a newly fine-tuned delta model $\delta^t(t)$ is added into the sequence and all the above $t$ delta models are unified.

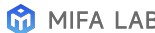

**Setting 2**: This setting is a transitional setting used in the proof. Unlike the normal continual learning process, in this setting, we only have $n$ delta models during all sessions. After the decoupling phase of each session $t$, we have $n$ delta models $\delta^1(t), \delta^2(t), \ldots, \delta^n(t)$ to be unified, and no newly fine-tuned delta model will be added in the sequence.

### G.2 PROOF

Before proving Theorem1, we prove Lemmas 1, Lemmas 2 and Corollary 1.

**Lemma 1.** *Let $X_1, X_2, \ldots, X_n$ be i.i.d. random variables. Then*

$$\mathbb{P}\big(X_n = \max\{X_1, \ldots, X_n\}\big) = \frac{1}{n}.$$

*Proof.* Assume that $X_1, X_2, \ldots, X_n$ are random variables with a common probability density function $f$ and cumulative distribution function $F$. We compute

$$\mathbb{P}\big(X_n = \max\{X_1, \ldots, X_n\}\big)$$
$$=\mathbb{P}\big(X_n > X_1, \ldots, X_n > X_{n-1}\big)$$
$$=\int_{-\infty}^{\infty} \mathbb{P}\big(x > X_1, \ldots, x > X_{n-1}\big) f(x) \, \mathrm{d}x.$$

Since $X_1, \ldots, X_{n-1}$ are i.i.d. with CDF $F$, we have

$$\mathbb{P}(X_1 < x, \ldots, X_{n-1} < x) = \big[F(x)\big]^{n-1}.$$

Hence

$$\mathbb{P}\big(X_n = \max\{X_1, \ldots, X_n\}\big) = \int_{-\infty}^{\infty} \big[F(x)\big]^{n-1} f(x) \, \mathrm{d}x$$
$$= \int_{-\infty}^{\infty} \big[F(x)\big]^{n-1} \, \mathrm{d}F(x)$$
$$= \int_{0}^{1} u^{n-1} \, \mathrm{d}u$$
$$= \frac{1}{n}.$$

This completes the proof. $\qquad\square$

**Lemma 2** (Sign Preservation of $\delta^i$)**.** *In setting 1, for a task $\delta^i$ in a continual learning session, the parameter at any position is guaranteed to preserve its sign. Specifically, if a position in $\delta^i(t')$ (denoted as $a^i(t')$) is positive (or negative) after an iteration, then $\forall t \geq t', a^i(t) \geq 0$ (or $\leq 0$). Moreover, if $a^i(t') = 0$, then $\forall t > t', a^i(t) = 0$.*

*Proof.* We observe that $M^i(t) = (\delta^i(t-1) \odot \delta(t))$. Therefore, if the signs remain consistent after an iteration, then $M^i(t) = 1$, and if the signs change, then $M^i(t) = 0$. As a result, positive (or negative) values in $\delta^i$ during the iteration process will either retain their signs or become zero. From the definition of $M^i(t)$, if $\delta^i(t-1) = 0$, it implies $M^i(t) = 0$, and thus $\delta^i(t) = 0$. Hence, the latter part of the lemma is also proved. $\qquad\square$

**Lemma 3** (Convergence of Iteration)**.** *In the setting 2, if the relative order of $\lambda^i(t)$ values remains unchanged and $\forall i \neq k, \forall t, \{j \mid M_j^i(t) = 1 \text{ and } M_j^k(t) = 1\} \neq \varnothing$, then these $n$ delta models will converge to a uniquely determined set of $n$ delta models.*

*Proof.* First, we show that $\forall i \in [1, \ldots, n], t > 0, M^i(1) = M^i(t)$.

Indeed, from the proof of Lemma 2, we know that the number of zeros in $M^i(t)$ does not decrease during iterations. Since no new delta models are introduced, the number of zeros in $M^i(t)$ cannot increase either. Thus, the sign of each position remains fixed during iterations.

Let $\delta^i_j(t)$ denote the value at the $j$-th position of the $i$-th delta model after $t$ iterations, and let $\phi(t)$ denote the delta model obtained by taking the absolute value of each position of $\delta(t)$. Note that $\delta^i(1)$ is obtained by scaling $\phi(1)$ with $\lambda^i(1)$ and setting certain positions to 0:

$$\phi_j(2) = \phi_j(1) \cdot \max\{\lambda^i(1) \mid M^i_j(1) = 1\}. \tag{1}$$

From Eq. (1), it follows that the delta model $\delta^i(1)$ with the largest $\lambda^i(1)$ contributes all its non-zero values to $\phi(2)$. Let $i_m$ denote the index of the $m$-th largest scaling factor $\lambda^i(1)$. Since the relative order of $\lambda^i(t)$ values remains unchanged, for $t > 1$, the real $i$ corresponding to $i_m$ remains in $\{1, 2, \ldots, n\}$.

For $t \geq 2$, we show that $\lambda^{i_1}(t) = 1$. By assumption, $\lambda^{i_1}(t) = \max\{\lambda^i(t) \mid i = 1, \ldots, n\}$ for all $t$. Consider the part of $\delta^{i_1}(t)$ where $M^{i_1}_k(t) = 1$. These positions remain unchanged in $\phi_j(t)$ during iterations. Hence, $\delta^{i_1}(t) = M^{i_1}(t) \odot \phi(t) = \delta^{i_1}(t+1)$, and $\lambda^{i_1}(t) = 1$.

Next, we consider $i_2$ and divide $\delta^{i_2}(t)$ into two parts. Let $x(t)$ denote the sum of absolute values in $\delta^{i_2}(t)$ at positions where $M^{i_1}(t) = 1$ during the $t$-th iteration, and let $y(t)$ represent the sum of absolute values at positions where $M^{i_1}(t) = 0$. By the definition of $\lambda^{i_2}(t)$, we know that $x(t) + y(t) = c$ for all $t$, where $c$ is a constant. During each iteration, the values in $y(t)$ are incorporated into $\phi(t)$. Let $s$ denote the sum of absolute values in $\delta^{i_1}(t)$ at positions where $M^{i_2}(t) = 1$, which is also a constant. Then, we have:

$$\begin{cases} \lambda^{i_2}(t) = \frac{x(t)+y(t)}{s+y(t)} = \frac{c}{s+y(t)}, \\ x(t+1) = \lambda^{i_2}(t) \cdot s = \frac{cs}{s+y(t)}, \\ y(t+1) = \lambda^{i_2}(t) \cdot y(t) = \frac{cy(t)}{s+y(t)}. \end{cases}$$

Since iterations do not alter the relative order of $\lambda^i(t)$, we have $\lambda^{i_2}(t) < \lambda^{i_1}(t) = 1$. Consequently, $0 < y(t+1) < y(t)$, which implies that both $x(t)$ and $y(t)$ converge, and $\delta^{i_2}(t)$ monotonically converges to a stable solution.

Using a similar analysis for $i_3, \ldots, i_n$, it can be shown that all $\delta^i(t)$ eventually converge to unique solutions. $\square$

**Corollary 1.** *Under the same conditions as Theorem 3, for each $i = 1, 2, \ldots, n$, and for sufficiently large $t$, the following inequality holds:*

$$\|\delta^i(t+1) - \delta^i(t)\|_1 < \|\delta^i(t) - \delta^i(t-1)\|_1.$$

**Proof of Theorem 1.**

*Proof.* Without loss of generality, assume that each position of $\delta^i(t)$ $\forall i \in \{1, \ldots, t\}$ is positive. Since we assume that $\delta(t)$ is bounded, and the absolute value at each position of $\delta^i(t)$ is less than $\delta(t)$, it follows that all $\delta^i(t)$ are bounded. Therefore, $\forall t$ and $i \in \{1, \ldots, t\}$, $\|\delta^i(t)\|_1$ is bounded, there exists a constant $S > 0$ such that

$$\|\delta^i(t)\|_1 < S, \quad \forall i \text{ and } t.$$

Denote by $\delta_*(t)$ the unified delta model selected in the iteration of $\{\delta^i(t)\}^{t-1}_{i=1}$, and let $\lambda^i_*(t)$ be the corresponding scaling factor for $\delta^i(t)$. The post-iteration results are $\{\delta^i_*(t+1)\}$.

According to Theorem 1, when iterating over the delta model set $\{\delta^i(t) \mid i \in \{1, \ldots, t\}\}$, for any positional dimension $p$, we have

$$\mathbb{P}\big(\delta_{*p}(t) \neq \delta_p(t)\big) = \mathbb{P}\big(\delta^t_p(t) = \max_{1 \leq i \leq t} \delta^i_p(t)\big) = \frac{1}{t}. \tag{2}$$

Since $\delta_p(t) = \max_{1 \leq i \leq t} \delta^i_p(t)$ and $\delta_{*p}(t) = \max_{1 \leq i \leq t-1} \delta^i_p(t)$,

let $f(t) = \mathbb{E}\big[\max_{1 \leq i \leq t} \delta^i_p(t)\big]$, and similarly $f(t-1) = \mathbb{E}\big[\max_{1 \leq i \leq t-1} \delta^i_p(t)\big]$, then

$$\lim_{t \to \infty} r(t) = 1,$$

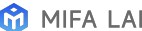

where $r(t) = \frac{f(t)}{f(t-1)}$.

Therefore,

$$\mathbb{E}\left[\frac{\delta_p(t)}{\delta_{*p}(t)}\right] = \frac{\mathbb{E}\left[\max\{\delta_p^i(t) \mid i = 1, \ldots, t\}\right]}{\mathbb{E}\left[\max\{\delta_p^i(t) \mid i = 1, \ldots, t-1\}\right]} = \frac{f(t)}{f(t-1)} = r(t). \tag{3}$$

Combining (2) and (3), it follows that

$$\begin{aligned}
\mathbb{E}\left[\frac{\|\delta(t)\|_1}{\|\delta_*(t)\|_1}\right] &= \mathbb{P}\left(\delta_p(t) = \delta_{*p}(t)\right) + \mathbb{P}\left(\delta_p(t) \neq \delta_{*p}(t)\right) \mathbb{E}\left[\frac{\delta_p(t)}{\delta_{*p}(t)}\right] \\
&= \frac{t-1}{t} + \frac{1}{t}\, r(t) \\
&= \frac{t-1+r(t)}{t},
\end{aligned}$$

and hence, for $i \in \{1, \ldots, t-1\}$,

$$\mathbb{E}\left[\frac{\lambda^i(t)}{\lambda_*^i(t)}\right] = \mathbb{E}\left[\frac{\|\delta^i(t)\|_1}{\|\delta(t)\|_1} \Big/ \frac{\|\delta^i(t)\|_1}{\|\delta_*(t)\|_1}\right] = \mathbb{E}\left[\frac{\|\delta_*(t)\|_1}{\|\delta(t)\|_1}\right] = \frac{t}{t-1+r(t)}.$$

Since

$$\|\delta^i(t+1) - \delta^i(t)\|_1 \leq \|\delta_*^i(t+1) - \delta^i(t)\|_1 + \|\delta^i(t+1) - \delta_*^i(t+1)\|_1,$$

and suppose the dimension of $\delta^i(t+1)$ is d, then

$$\begin{aligned}
&\mathbb{E}\left[\|\delta^i(t+1) - \delta_*^i(t+1)\|_1\right] \\
&= \mathbb{P}\left(\delta_{*p}(t) = \delta_p(t)\right) \left|\lambda^i(t) - \lambda_*^i(t)\right| \mathbb{E}\left[\|\delta(t)\|_1\right] + \mathbb{P}\left(\delta_{*p}(t) \neq \delta_p(t)\right) \mathbb{E}\left[\|\delta^i(t+1) - \delta_*^i(t+1)\|_1\right] \\
&= \frac{t-1}{t} \frac{r(t)-1}{t} \lambda^i(t) \mathbb{E}\left[\|\delta(t)\|_1\right] + \frac{1}{t} \mathbb{E}\left[\|\delta_p^i(t+1) - \delta_{*p}^i(t+1)\|_1 \mid \delta_{*p}(t) \neq \delta_p(t)\right] \\
&\leq \frac{(t-1)(r(t)-1)}{t^2} \mathbb{E}\left[\|\delta(t)\|_1\right] + \frac{d}{t} \mathbb{E}\left[\left|\delta_p^i(t+1) - \delta_{*p}^i(t+1)\right| \mid \delta_{*p}(t) \neq \delta_p(t)\right] \\
&= \frac{(t-1)(r(t)-1)}{t^2} \mathbb{E}\left[\|\delta(t)\|_1\right] + \frac{d}{t}\left(\mathbb{E}\left[\lambda^i(t)\delta_p(t)\right] - \mathbb{E}\left[\lambda_*^i(t)\delta_{*p}(t)\right]\right) \\
&\leq \frac{(t-1)(r(t)-1)}{t^2} \mathbb{E}\left[\|\delta(t)\|_1\right] + \frac{d}{t}\left(\frac{t}{t-1+r(t)}\, r(t) - 1\right) \frac{\mathbb{E}\left[\|\delta_*(t)\|_1\right]}{d} \\
&\leq \frac{(t-1)(r(t)-1)}{t^2} S + \frac{(t-1)(r(t)-1)}{t(t-1+r(t))} S.
\end{aligned}$$

Hence

$$\lim_{t \to \infty} \mathbb{E}\left[\|\delta^i(t+1) - \delta_*^i(t+1)\|_1\right] = 0.$$

Moreover, by Corollary 1, it follows that

$$\mathbb{E}\left[\|\delta_*^i(t+1) - \delta^i(t)\|_1\right] < \mathbb{E}\left[\|\delta^i(t) - \delta^i(t-1)\|_1\right].$$

Therefore, for $i \in \{1, \ldots, t-1\}$, we conclude

$$\lim_{t \to \infty}\left(\mathbb{E}\left[\|\delta^i(t+1) - \delta^i(t)\|_1\right] - \mathbb{E}\left[\|\delta^i(t) - \delta^i(t-1)\|_1\right]\right) \leq 0.$$

$\qquad\qquad\qquad\qquad\qquad\qquad\qquad\qquad\qquad\qquad\qquad\qquad\qquad\qquad\qquad\qquad\qquad\quad\square$

This finishes the proof.

## H STORAGE ANALYSIS

The masks align the unified delta model's direction with that of each delta model, and the rescalers ensure that the unified model's parameter magnitude matches that of each delta model. Although the masks share the structure of the delta model, their binary nature ensures they require much less storage than the delta models, and the rescalers are $t$ scalars whose storage is negligible. We compare the storage size of model parameter files saved in Python between our method and Individual FT,

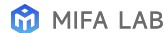

The results of Individual FT represent the performance of using a fully fine-tuned model, trained independently on each task based on the pre-trained VLM, for prediction.

First, we look at the comparison in the full fine-tuning scenario. After training all 11 tasks, the storage size for ConDU is as follows: Clip Model (570.86 MB) + Unified Delta Model (570.86 MB) + Masks (196.20 MB) + Rescalers (747 KB) = 1377.92 MB. In contrast, Individual FT's storage size is Task-specific Model (570.86 MB) × 11 = 6279.46 MB, saving a total of 4901.54 MB in storage.

Next, we compare in the LoRA (Rank=64) scenario. After training all 11 tasks, the storage size for ConDU is as follows: Clip Model (570.86 MB) + Unified Delta Model of LoRA (37.53 MB) + Masks of LoRA (12.89 MB) + Rescalers (747 KB) = 621.28 MB. For Individual FT, the storage size is Clip Model (570.86 MB) + LoRA (37.53 MB) × 11 = 983.51 MB, saving a total of 362.23 MB in storage.

As shown, our method significantly alleviates the excessive storage requirement of Individual FT, and the storage reduction is more pronounced as the proportion of fine-tunable parameters increases and the number of tasks grows.

## I  TRAINING AND INFERENCE TIME

We compare the training and inference time between ConDU and a SOTA method ZSCL (Zheng et al., 2023). Both methods are evaluated on a single GPU with computational power equivalent to an GeForce 4090. "s" denotes "seconds" here.

**Training Time.**  To demonstrate concrete performance, we analyze the SUN397 dataset from the last training session. For a fair comparison, we maintain the same settings, where the model is trained for 1000 iterations across all methods. The training pipeline of our method comprises four distinct phases: decoupling the unified model, tuning individually, computing prototypes, and unifying models.

For ConDU (LoRA), the total training time of this session is decoupling (0.64s) + unifying (3.72s) + tuning (401.72s) + computing proto (303.63s) = 709.71s. In constrast, the total training time of ConDU (FT), ZSCL and Continual Fine-tuning is 738.91s, 1504s and 443s. According to the results, ConDU does not significantly increase training time compared to Continual Fine-tuning and saves approximately 52% time relative to ZSCL.

**Inference Time.**  At the inference stage, if a test sample comes from a known task with a provided task ID, the corresponding task-specific model is selected for prediction, making the inference time equivalent to that of a single model. When dealing with test samples from unseen tasks or without a task ID, our ConDU (LoRA) method involves two phases: task-specific model selection based on computing cosine similarity, and aggregating the predictions of the selected models. We take the SUN397 dataset in the last session as an example. The model selection phase takes 0.27s. Since the forward propagation of the selected task-specific models can be computed in parallel, the whole aggregating phase takes nearly the same time as the prediction of a single model, which is just 30.22s. In contrast, ZSCL takes 29.88s at the inference stage. The additional 1.58s consumed by our method due to extra algorithmic steps accounts for only 5.16% of the total inference time. Since ConDU (FT) requires higher computational resources for parallel inference, we recommend using ConDU (LoRA) for scenarios where inference speed is a concern.

## J  COMPREHENSIVE COMPARISON BETWEEN CONDU AND INDIVIDUAL FT

We now provide an integrated comparison between Individual FT (i.e., training and storing a separate expert model for each task without unification) and ConDU in Table 12.

"IF" denotes Individual FT. "CD" denotes ConDU. The results show that:

- Accuracy (on standard MTIL, using the "Last" metric): Individual FT achieves slightly better accuracy than ConDU.

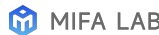

Table 12: Comprehensive Comparison between ConDU and Individual FT.

| Method | Aircraft | Caltech101 | CIFAR100 | DTD | EuroSAT | Flowers | Food | MNIST | OxfordPet | Cars | SUN397 | *Avg* | *FMS (MB)* | *IT (s)* |
|---|---|---|---|---|---|---|---|---|---|---|---|---|---|---|
| IF (FT) | 62.0 | 95.1 | 89.6 | 79.5 | 98.9 | 97.5 | 92.7 | 99.6 | 94.7 | 89.6 | 81.8 | 89.2 | 6279.46 | 28.34 |
| IF (LoRA) | 59.5 | 97.3 | 89.0 | 79.9 | 98.6 | 97.7 | 92.8 | 99.4 | 94.3 | 90.5 | 81.9 | 89.2 | 983.51 | 29.95 |
| CD (FT) | 58.6 | 93.7 | 86.6 | 76.1 | 98.2 | 93.4 | 91.9 | 99.6 | 94.8 | 84.9 | 80.5 | 87.1 | 1377.92 | 28.75 |
| CD (LoRA) | 48.9 | 95.2 | 87.8 | 78.5 | 96.3 | 95.2 | 91.7 | 97.6 | 93.0 | 85.3 | 78.8 | 86.2 | 621.28 | 30.22 |

Table 13: Performance across sessions on different datasets.

| | Aircraft | Caltech101 | CIFAR100 | DTD | EuroSAT | Flowers | Food | MNIST | OxfordPet | Cars | SUN397 | *Average* |
|---|---|---|---|---|---|---|---|---|---|---|---|---|
| Session 1 | 61.3 | | | | | | | | | | | 61.3 |
| Session 2 | 60.5 | 94.6 | | | | | | | | | | 77.6 |
| Session 3 | 59.8 | 93.9 | 87.9 | | | | | | | | | 80.5 |
| Session 4 | 60.0 | 94.0 | 87.5 | 77.1 | | | | | | | | 79.7 |
| Session 5 | 59.9 | 93.9 | 87.2 | 76.7 | 98.4 | | | | | | | 83.2 |
| Session 6 | 59.8 | 93.9 | 87.2 | 76.8 | 98.3 | 95.0 | | | | | | 85.2 |
| Session 7 | 59.5 | 93.9 | 87.1 | 76.5 | 98.3 | 94.5 | 92.3 | | | | | 86.0 |
| Session 8 | 58.7 | 93.8 | 86.5 | 75.9 | 98.1 | 93.4 | 92.0 | 99.6 | | | | 87.3 |
| Session 9 | 58.8 | 93.8 | 86.5 | 76.0 | 98.2 | 93.4 | 91.9 | 99.6 | 94.9 | | | 88.1 |
| Session 10 | 58.5 | 93.8 | 86.5 | 76.0 | 98.2 | 93.4 | 91.9 | 99.6 | 94.8 | 85.7 | | 87.8 |
| Session 11 | 58.6 | 93.7 | 86.6 | 76.1 | 98.2 | 93.4 | 91.9 | 99.6 | 94.8 | 84.9 | 80.5 | 87.1 |

- Storage: "FMS" (Final Model Size) measures the size of the model after all sessions are completed. Individual FT incurs significantly higher storage costs than ConDU, as all expert models are stored independently.
- Inference Time: "IT" denotes "Inference Time". While ConDU introduces a lightweight expert-weight computation step, overall inference latency is nearly unchanged. The inference time here refers to the inference time of the last session on the SUN397 test set.
- Generality: Importantly, Individual FT does not support zero-shot inference or task-agnostic settings, limiting its applicability. Thus, the comparsion only includes the "Last" metric on standard MTIL.

## K  ACCURACY DURING SESSIONS

Table 13 shows the test accuracy across all tasks at the end of each session. Based on our experimental observations, the performance of all previous tasks decreases more after MNIST session than other sessions due to the significant differences between the handwritten digit recognition task (MNIST) and other tasks that focus on object classification. However, the magnitude of the drop remains small, demonstrating ConDU's strong robustness even when encountering highly OOD tasks.

## L  INTRODUCTION OF LoRA

Below we revisit LoRA, in which $g$ is the module that PEFT attached to, $\mathbf{e}$ and $\mathbf{h}$ are input and output of the original $g$ and $\mathbf{h}'$ is output of $g$ attached with PEFT.

**LoRA** (Hu et al., 2021) assumes the change of parameters is in a low-rank space when tuning the pre-trained model on a downstream task. For a linear layer with weight $\mathbf{W} \in \mathbb{R}^{d \times d'}$, the weight

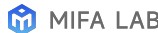 MIFA LAB

updates $\Delta \mathbf{W}$ can be decomposed into the multiplication of two small matrices:

$$\Delta \mathbf{W} = \mathbf{W}_{down} \mathbf{W}_{up},$$

where $\mathbf{W}_{down} \in \mathbb{R}^{d \times r}$ and $\mathbf{W}_{up} \in \mathbb{R}^{r \times d'}$. For the convolution layer, the updates can be reshaped into the kernel shape. Finally, LoRA modifies the forward pass of the adapted layer into the following form:

$$\mathbf{h}' = \mathbf{h} + \mathbf{e} * (\mathbf{W}_{down} \mathbf{W}_{up}),$$

where $*$ is matrix multiplication or convolution operation, the bias and reshape operation are omitted for conciseness. Since LoRA adapts the weight of $g$, the weight updates can be merged into $g$ to reduce the inference latency.

## M  THE USE OF LARGE LANGUAGE MODELS (LLMS)

The LLMs are used only to help polishing writing in this work.

