# OpenReview forum: "Enhanced Continual Learning of Vision-Language Models with Model Fusion"
_ICLR.cc/2026/Conference — ICLR 2026 Poster_

### Official Review · Reviewer_j5xv · 2025-10-17

**Soundness:** 4
**Presentation:** 2
**Contribution:** 3
**Rating:** 6
**Confidence:** 3

**Summary:**

This paper introduces ConDU (Continual Decoupling–Unifying), a novel framework for continual learning (CL) of vision-language models (VLMs) such as CLIP. The core idea is to employ model fusion—traditionally used for combining independently fine-tuned models—to enable continual learning without catastrophic forgetting or reliance on replay data. Extensive experiments on MTIL, task-agnostic MTIL, and few-shot MTIL benchmarks show that ConDU outperforms existing methods by up to 2% in average accuracy while maintaining or improving zero-shot transfer performance.

**Strengths:**

1. Strong empirical gains across multiple benchmarks and settings.
2. Novel use of model fusion for continual learning of VLMs.
3. Fast inference and training efficiency due to training-free decoupling/unifying steps.

**Weaknesses:**

While the technical contribution is solid, the presentation quality should be improved to make the paper easier to follow. If these issues are addressed, I would be inclined to raise my evaluation.
Key suggestions:
1.	Figure clarity.
The text in figures (e.g., Fig. 1, 3, 4) is too small to read. Redrawing them with larger labels and more space would improve readability.
2.	Figure redundancy.
Fig. 1 is not clearly explained and highly overlaps with Fig. 3; consider merging the key information into Fig. 3 and removing Fig. 1 to streamline the narrative.
3.	Structure of the paper.
The introduction currently contains too much methodological detail. The core method description should be moved entirely into Section 4 (“ConDU”), leaving the introduction focused on motivation and contribution.
4.	Language and phrasing.
Several passages could benefit from linguistic polishing for clarity and conciseness. Improving transitions and consistent terminology would greatly enhance readability.

**Questions:**

1.	Scalability.
How does ConDU behave as the number of tasks increases significantly (e.g., >20)? Does the unifying process saturate, or does performance plateau as the unified delta model grows?
2.	Memory usage.
Since ConDU stores task triggers and prototype sets, how does the memory footprint scale with the number of tasks? Could the method benefit from pruning or compressing old triggers to maintain constant memory?

---

> ### Author Response · Authors · 2025-11-21
> **Response to Reviewer j5xv (1/2)**
>
> **W1:** While the technical contribution is solid, the presentation quality should be improved to make the paper easier to follow. If these issues are addressed, I would be inclined to raise my evaluation. Key suggestions: 1. Figure clarity. The text in figures (e.g., Fig. 1, 3, 4) is too small to read. Redrawing them with larger labels and more space would improve readability. 2. Figure redundancy. Fig. 1 is not clearly explained and highly overlaps with Fig. 3; consider merging the key information into Fig. 3 and removing Fig. 1 to streamline the narrative. 3. Structure of the paper. The introduction currently contains too much methodological detail. The core method description should be moved entirely into Section 4 (“ConDU”), leaving the introduction focused on motivation and contribution. 4. Language and phrasing. Several passages could benefit from linguistic polishing for clarity and conciseness. Improving transitions and consistent terminology would greatly enhance readability.
>
> **A:** Thank you very much for your helpful suggestions regarding the presentation quality of our paper. **Based on your comments, we have made the following improvements in the revised version**:
>
> 1. **We have merged the original Figure 1 and Figure 3 into a single figure** (now presented as Figure 1 in revised version) to increase information density and make the illustration easier to understand. The prototype computation procedure is separated into its own figure (Figure 3 in revised version).
>
> 2. **We have enlarged all figure labels, increased figure size**, and expanded spacing to enhance readability, with particular improvements made to Figures 1–4 of revised version.
>
> 3. **We have reduced the methodological descriptions in the introduction** to a single sentence and moved the core method explanation entirely to Section 4, allowing the introduction to focus on motivation and contributions. You may refer to Lines 66–73 of revised version to evaluate whether this restructuring meets the intended goal of improving the introduction.
>
> 4. **We have unified terminology** throughout the paper to improve readability. These updates can be found in Lines 21 23 59 75 77 161 206 235 260 262 268 270 287 474 479 483 523 (revised version). The core terms in the revised version now include:
> - **Pre-trained VLM**: The original vision–language model before any fine-tuning.
> - **Task Expert**: A model obtained by fine-tuning the pre-trained VLM on a specific task.
> - **Delta Model**: The parameter difference between a task expert and the pre-trained VLM for that task.
> - **Unified Delta Model**: A set of parameters obtained by unifying the delta models of all tasks.
> - **Unified Model**: The pre-trained VLM combined with the unified delta model.
> - **Task Trigger**: A mask–scalar pair computed from the differences between a task’s delta model and the unified delta model, used during the decoupling step to reconstruct the corresponding task expert.
>
> 5. **We have further polished the writing in multiple sections** (mainly in Section 1,3,4) to make the paragraphs more concise and clear. All modified content is highlighted in blue in revised version. If you notice additional paragraphs that could benefit from further refinement, we would be grateful for your guidance and will be happy to improve them.
>
> If you have any additional suggestions regarding the presentation of the revised manuscript after reviewing it, we would be very grateful to hear them and will promptly submit an improved version accordingly.
>
> ---

---

> > ### Author Response · Authors · 2025-11-21
> > **Response to Reviewer j5xv (2/2)**
> >
> > **Q1:** Scalability. How does ConDU behave as the number of tasks increases significantly (e.g., >20)? Does the unifying process saturate, or does performance plateau as the unified delta model grows?
> >
> > **A:** This is an excellent exploratory question—namely, whether ConDU remains effective when the number of tasks increases further. To investigate this, we conducted additional experiments. We split each task in ConDU into three smaller tasks, thereby **constructing a longer task sequence (33 tasks)**, named Split MTIL. We evaluated our method and three strongest baselines on Split MTIL using exactly the same evaluation protocol as in the original paper. The results show that **even with a much longer task sequence, our method still maintains clear advantages, which indirectly suggests that the unifying process has not saturated**.
> >
> > |               | Transfer | Average  | Last |
> > | ------------- | -------- | ---- | ---- |
> > | ConDU (FT) | 81.3     | 87.0 | 90.9 |
> > | ZSCL          | 78.3     | 83.6 | 89.0 |
> > | Primal-rail   | 56.9     | 68.4 | 80.5 |
> > | Dual-rail     | 57.2     | 68.6 | 81.5 |
> >
> > Regarding whether the performance tends to plateau, please refer to Lines 1334–1339 of revised version. We observe that as the number of tasks increases, once a task expert has been reconstructed beyond a certain number of times, the performance degradation becomes small. In addition, the theoretical analysis in Lines 481–534 also states that under certain assumptions, as the task sequence grows indefinitely, the parameter changes of each individual task expert become progressively smaller.
> >
> > ---
> >
> > **Q2:** Memory usage. Since ConDU stores task triggers and prototype sets, how does the memory footprint scale with the number of tasks? Could the method benefit from pruning or compressing old triggers to maintain constant memory?
> >
> > **A:** First, regarding “the memory footprint”, this is indeed an important issue. The discussion in Lines 1234–1254 of revised version provides a detailed analysis of this point. We have compiled statistics on the composition of ConDU's memory storage after training all 11 tasks (excluding the pre-trained CLIP model's storage 570.86 MB) and provided the storage of Individual FT (training individual expert for each task) for comparison.
> > - Full fine-tuning scenario: ConDU's storage (770.15 MB) = Unified Delta Model (570.86 MB) + Task Triggers (196.94 MB) + Prototype Sets (2.35 MB). Individual FT’s storage (6279.46 MB) = Task Expert (570.86 MB) * 11.
> > - LoRA (Rank=64) scenario: ConDU's storage (53.51 MB) = Unified Delta Model (37.53 MB) + Task Triggers (13.63 MB) + Prototype Sets (2.35 MB). Individual FT's storage (412.83 MB) = Task Expert (37.53 MB) * 11.
> >
> > From the statistics, **task triggers account for 14.6% (FFT) and 2.1% (LoRA) of the total storage, while prototype sets account for 0.1% (FFT) and 0.3% (LoRA)**.
> >
> > Second, as mentioned in first point, **the memory fraction contributed by task triggers is small**, so we have not considered compressing task triggers at this stage. Of course, when the task sequence becomes extremely long, task triggers may take up a larger portion of memory—for example, in the LoRA fine-tuning scenario, **once the number of tasks exceeds 500, task triggers may exceed 50% of the storage**. On one hand, such scenarios are extremely rare in practical applications; on the other hand, existing methods also fail under such extreme settings, so this potentially forms a new research problem.
> >
> > Finally, if we truly **encounter such extremely long sequences** as described in second point, **it is indeed feasible to modify ConDU to adapt to this situation**. One option is to compress task triggers. For instance, we could fix $K$ binary vectors with the same dimensionality as the task triggers to form a dictionary, and approximate each task trigger as a linear combination of these $K$ binary vectors. Although this introduces some information loss, given the highly sparse nature of task triggers, such an approach is likely to remain effective. We plan to explore this direction in future work.
> >
> > ---

---

### Official Review · Reviewer_1yyf · 2025-10-25

**Soundness:** 3
**Presentation:** 3
**Contribution:** 2
**Rating:** 6
**Confidence:** 5

**Summary:**

This paper proposes ConDU, a continual learning framework for VLMs that introduces training-free decoupling–unifying model fusion. A single unified model is maintained; per-task “triggers” (sign mask + rescaler) reconstruct task-specific deltas, and a prototype-based semantic aggregator combines multiple reconstructed experts for zero-shot or unknown-task inference. ConDU supports both full fine-tuning and LoRA. On MTIL (plus task-agnostic and few-shot variants), it improves Average by up to ~2% over SOTA while preserving/boosting zero-shot transfer, with low fusion overhead (~1% of fine-tuning time).

**Strengths:**

**1. Model-fusion continual learning, broadly compatible.** ConDU formalizes a decouple–unify routine with task triggers, works with full FT and PEFT, and keeps only one deployable model plus small metadata—no replay or reference dataset.

**2. Zero-shot/unknown-task inference.** Prototype-weighted aggregation of reconstructed experts enables robust prediction without task IDs, aligning with VLM zero-shot goals.

**3. Consistent empirical gains and efficiency.** Across MTIL variants, ConDU raises Transfer/Average/Last and reports modest storage/time costs; decoupling/unifying are training-free and ~1% of tuning.

**Weaknesses:**

**1. Limited novelty.** The contribution largely integrates known model-merging ideas into a continual pipeline; the unify rule (signed extreme per parameter) and trigger design, while clever, are incremental versus established fusion schemes (Fisher, TIES, EMR). Stronger positioning against these would clarify originality.

**2. Lacking sufficient ablation study.** Beyond an insensitivity check on the top-K selector, there is little analysis of design choices: unify rule vs. Fisher/TIES/EMR, trigger rescaling, mask binarization, prototype variants, or alternative gating (learned vs. cosine). Such ablations would substantiate the mechanism-level claims.

**3. Benchmark scope.** Results focus on MTIL and its task-agnostic/few-shot variants (11 domains, classification); broader VLM tasks (detection, OCR, medical, video, captions) aren’t covered, limiting external validity.

**4. Assumptions/theory.** Convergence analysis assumes identical parameter signs and i.i.d. deltas—conditions unlikely across heterogeneous tasks—so theoretical guarantees may not reflect practice.

**5. Aggregation sensitivity.** Zero-shot prediction hinges on prototypes from the pre-trained VLM; domain shifts or text prompt bias could mis-weight experts, and calibration is not discussed.

**6. Efficiency reporting.** While fusion is fast and memory-light, end-to-end latency under multi-expert aggregation and edge constraints is only briefly discussed; more granular profiles would aid adoption.

**Questions:**

1. How would performance change if the unify step used Fisher/TIES/EMR-style merging or soft masks/rescalers; which component (masking, rescale, signed-extrema) drives gains?

2. Can learned routers or temperature-calibrated cosine improve task weighting, and how robust is aggregation to prompt wording or domain shift in the prototype bank?

3. What happens when the theorem’s sign/i.i.d. assumptions are violated—e.g., conflicting signs across tasks—and can relaxed bounds be established empirically or analytically?

4. What are end-to-end latency/throughput and memory footprints for K-expert aggregation on commodity GPUs/edge devices, and how does this compare to MoE or per-task checkpoints?

---

> ### Author Response · Authors · 2025-11-21
> **Response to Reviewer 1yyf (1/6)**
>
> **W1:** Limited novelty. The contribution largely integrates known model-merging ideas into a continual pipeline; the unify rule (signed extreme per parameter) and trigger design, while clever, are incremental versus established fusion schemes (Fisher, TIES, EMR). Stronger positioning against these would clarify originality.
>
> **A:** First, our work is not merely a straightforward combination of model merging and continual learning. Rather, it is **the first to propose model merging itself as a technical route for continual learning of VLMs**. Unlike existing approaches, this route naturally accommodates full fine-tuning, partial fine-tuning, and parameter-efficient fine-tuning, thereby opening a new direction for this domain.
>
> Second, we would like to emphasize that existing model-merging techniques cannot be directly applied to our setting. **Prior methods are designed for a single, one-time fusion** of multiple models; their fused outputs cannot be merged again, **making them unsuitable for continual learning, where multi-stage merging is required**. Our contribution is to introduce an entirely new fusion strategy that supports repeated merging.
>
> For example, Fisher Merging requires computing a Fisher weight for each parameter of each model using its training data. However, the merged model is not obtained through training and thus cannot produce Fisher weights; moreover, in continual learning, the training data of previous tasks is no longer available. As a result, Fisher Merging cannot perform another round of merging. In TIES-Merging, once the models have been averaged, the original sign conflicts between them are no longer preserved, making the method unsuitable for scenarios that require repeated merging. In EMR-Merging, the fusion output is represented as a combination of parameters and masks, whose form differs from the original models, making further merging impossible.
> In contrast, ConDU is specifically designed for continual learning: it enables fusion between the previously merged result and the newly fine-tuned model without requiring any past training data. This setting is not handled—and has not been explored—by Fisher, TIES, or EMR.
>
> Third, our method incorporates several designs tailored to continual learning. For each session, we store the differences among task experts in the task triggers, while retaining their shared information in the unified delta model. Each dimension of the unified delta model aggregates the corresponding dimensions across delta models. These designs aim to ensure that each task expert remains as consistent as possible after reconstruction, aligning with the stability and plasticity goals in continual learning. This consistency is also verified in the paper (revised version) through quantitative evidence (Lines 1334–1339), t-SNE visualizations (Lines 473–480), and theoretical analysis (Lines 481–534).
>
> ---

---

> > ### Author Response · Authors · 2025-11-21
> > **Response to Reviewer 1yyf (2/6)**
> >
> > **W2:** Lacking sufficient ablation study. Beyond an insensitivity check on the top-K selector, there is little analysis of design choices: unify rule vs. Fisher/TIES/EMR, trigger rescaling, mask binarization, prototype variants, or alternative gating (learned vs. cosine). Such ablations would substantiate the mechanism-level claims.
> >
> > **Q1:** How would performance change if the unify step used Fisher/TIES/EMR-style merging or soft masks/rescalers; which component (masking, rescale, signed-extrema) drives gains?
> >
> > **A:** Thank you for the thoughtful question. Our method is built as an integrated design to preserve per-task expert consistency after reconstruction. Removing any component breaks this overall mechanism; the method is not a composition of independent, swappable modules. Nonetheless, we provide detailed ablations and reasoning for each component:
> >
> > 1. **Unify rule**: As discussed in our first response, Fisher/TIES/EMR are designed for one-time model merging. Their merged model cannot be repeatedly merged again, which makes them incompatible with continual learning where iterative merging is essential. Our unify rule is specifically constructed for multi-step continual fusion; therefore, **directly replacing it with Fisher/TIES/EMR is infeasible**.
> >
> > 2. **Trigger rescaling**: While the review does not specify the exact ablation desired, we interpret the request as asking whether rescaling materially affects performance. Importantly, **without the rescaler, the reconstructed task experts will produce features with significantly mismatched magnitudes compared to the original task experts**. This motivates the inclusion of the rescaling module. We added an explicit ablation comparing ConDU (FT) with its no-rescaler variant. The variant performs substantially worse, confirming the necessity of rescaling. This ablation result is now included in the revised version (Lines 432–437).
> >
> > **Transfer**
> > |                    | AirCraft | Caltech101 | CIFAR100 | DTD  | EuroSAT | Flowers | Food | MNIST | OxfordPEt | Cars | SUN397 | Avg  |
> > | ------------------ | -------- | ---------- | -------- | ---- | ------- | ------- | ---- | ----- | --------- | ---- | ------ | ---- |
> > | ConDU              | —        | 88.1       | 68.9     | 46.4 | 57.0    | 71.3    | 88.7 | 65.5  | 89.3      | 65.0 | 67.8   | 70.8 |
> > | ConDU w/o rescaler | —        | 88.2       | 68.7     | 46.1 | 56.5    | 71.4    | 88.7 | 63.8  | 89.1      | 64.9 | 66.7   | 70.4 |
> >
> > **Avg**
> > |                    | AirCraft | Caltech101 | CIFAR100 | DTD  | EuroSAT | Flowers | Food | MNIST | OxfordPEt | Cars | SUN397 | Avg  |
> > | ------------------ | -------- | ---------- | -------- | ---- | ------- | ------- | ---- | ----- | --------- | ---- | ------ | ---- |
> > | ConDU              | 59.6     | 93.4       | 83.7     | 68.1 | 83.4    | 83.7    | 90.1 | 76.7  | 90.6      | 68.6 | 68.6   | 78.8 |
> > | ConDU w/o rescaler | 59.2     | 90.8       | 81.5     | 62.2 | 82.5    | 52.0    | 88.8 | 76.8  | 87.8      | 66.7 | 67.8   | 74.2 |
> >
> > **Last**
> > |                    | AirCraft | Caltech101 | CIFAR100 | DTD  | EuroSAT | Flowers | Food | MNIST | OxfordPEt | Cars | SUN397 | Avg  |
> > | ------------------ | -------- | ---------- | -------- | ---- | ------- | ------- | ---- | ----- | --------- | ---- | ------ | ---- |
> > | ConDU              | 58.6     | 93.7       | 86.6     | 76.1 | 98.2    | 93.4    | 91.9 | 99.6  | 94.8      | 84.9 | 80.5   | 87.1 |
> > | ConDU w/o rescaler | 57.8     | 89.9       | 83.1     | 66.2 | 96.9    | 30.3    | 88.7 | 99.6  | 84.4      | 74.3 | 77.1   | 77.1 |
> >
> > 3. **Mask binarization**: All masks are algorithmically computed per session and are not tunable hyperparameters. Hence, there is no meaningful ablation to perform on "masking".
> >
> > 4. **Signed-extrema**: Signed-extrema is co-designed with task triggers to ensure the unified delta model preserves as much information as possible and to keep reconstructed experts close to their originals. There are no reasonable alternative strategies serving the same purpose, so an ablation here is not well-defined.

---

> > > ### Author Response · Authors · 2025-11-21
> > > **Response to Reviewer 1yyf (3/6)**
> > >
> > > 5. **Prototype variants**: Although the review does not specify which “prototype variants” should be ablated, we interpret this as a question about whether prototypes should be computed from PTM features or from task-expert features. We added an additional study comparing prototype–sample similarity computed using PTM features (our default), and features extracted from each task’s expert model. In the expert-based variant, each class prototype is recomputed using its corresponding task expert; at inference, similarity is also computed in that expert’s feature space. Keeping all other components fixed, **the PTM-based strategy consistently outperforms the expert-based one on MTIL benchmark**, indicating that shared PTM representations provide a more reliable cross-task similarity basis. We only evaluated the Transfer and Average metrics, since the Last metric is not effected by the aggregation strategy. This ablation result is now included in the revised version (Lines 424–431).
> > >
> > > **Transfer**
> > > |                                      | AirCraft | Caltech101 | CIFAR100 | DTD  | EuroSAT | Flowers | Food | MNIST | OxfordPEt | Cars | SUN397 | Avg  |
> > > | ------------------------------------ | -------- | ---------- | -------- | ---- | ------- | ------- | ---- | ----- | --------- | ---- | ------ | ---- |
> > > | Expert-based | —        | 88.2       | 68.9     | 40.3 | 49.1    | 69.7    | 86.4 | 62.9  | 84.6      | 59.7 | 67.4   | 67.7 |
> > > | PTM-based     | —        | 88.1       | 68.9     | 46.4 | 57.0    | 71.3    | 88.7 | 65.5  | 89.3      | 65.0 | 67.8   | 70.8 |
> > >
> > > **Average**
> > > |                                      | AirCraft | Caltech101 | CIFAR100 | DTD  | EuroSAT | Flowers | Food | MNIST | OxfordPEt | Cars | SUN397 | Avg  |
> > > | ------------------------------------ | -------- | ---------- | -------- | ---- | ------- | ------- | ---- | ----- | --------- | ---- | ------ | ---- |
> > > | Expert-based | 59.6     | 93.4       | 83.7     | 67.5 | 81.7    | 82.7    | 89.1 | 77.0  | 88.2      | 65.1 | 68.3   | 77.8 |
> > > | PTM-based     | 59.6     | 93.4       | 83.7     | 68.1 | 83.4    | 83.7    | 90.1 | 76.7  | 90.6      | 68.6 | 68.6   | 78.8 |
> > >
> > > We hope these clarifications and additional results address the reviewer’s concerns regarding ablations.
> > >
> > > ---
> > >
> > > **W3:** Benchmark scope. Results focus on MTIL and its task-agnostic/few-shot variants (11 domains, classification); broader VLM tasks (detection, OCR, medical, video, captions) aren’t covered, limiting external validity.
> > >
> > > **A:** Our work focuses on image–text matching VLMs, which is also where the majority of recent continual learning of VLMs research has concentrated. **To ensure fairness and comparability, our backbone, benchmark choice, and evaluation protocol follow the most recent works** from 2024–2025 (see references [1-5]), which all adopt the same MTIL-style setting.
> > >
> > > That said, we believe ConDU is not limited to image–text matching. The method is model-agnostic and can be adapted to other VLM tasks by replacing the backbone accordingly. Extending ConDU to broader tasks such as detection, OCR, medical imaging, video, or captioning is an interesting direction, and we plan to explore these in future work.
> > >
> > > [1] Hongsheng Zhang, Zhong Ji, Jingren Liu, Yanwei Pang, and Jungong Han. Multi-stage knowledge integration of vision-language models for continual learning. arXiv preprint arXiv:2411.06764, 2024.
> > >
> > > [2] Jiazuo Yu, Yunzhi Zhuge, Lu Zhang, Ping Hu, Dong Wang, Huchuan Lu, and You He. Boosting continual learning of vision-language models via mixture-of-experts adapters. In Proceedings of the IEEE/CVF Conference on Computer Vision and Pattern Recognition, pp. 23219–23230, 2024.
> > >
> > > [3] Yicheng Xu, Yuxin Chen, Jiahao Nie, Yusong Wang, Huiping Zhuang, and Manabu Okumura. Advancing cross-domain discriminability in continual learning of vision-language models. arXiv preprint arXiv:2406.18868, 2024.
> > >
> > > [4] Zangwei Zheng, Mingyuan Ma, Kai Wang, Ziheng Qin, Xiangyu Yue, and Yang You. Preventing zero-shot transfer degradation in continual learning of vision-language models. In Proceedings of the IEEE/CVF International Conference on Computer Vision, pp. 19125–19136, 2023.
> > >
> > > [5] Yu-Chu Yu, Chi-Pin Huang, Jr-Jen Chen, Kai-Po Chang, Yung-Hsuan Lai, Fu-En Yang, and YuChiang Frank Wang. Select and distill: Selective dual-teacher knowledge transfer for continual learning on vision-language models. In European Conference on Computer Vision, pp. 219–236. Springer, 2025.
> > >
> > > ---

---

> > > > ### Author Response · Authors · 2025-11-21
> > > > **Response to Reviewer 1yyf (4/6)**
> > > >
> > > > **W4:** Assumptions/theory. Convergence analysis assumes identical parameter signs and i.i.d. deltas—conditions unlikely across heterogeneous tasks—so theoretical guarantees may not reflect practice.
> > > >
> > > > **Q3:** What happens when the theorem’s sign/i.i.d. assumptions are violated—e.g., conflicting signs across tasks—and can relaxed bounds be established empirically or analytically?
> > > >
> > > > **A:** Thank you for the thoughtful comment.
> > > >
> > > > First, **the “i.i.d. deltas” assumption is reasonable**. Each delta model is the parameter difference between the PTM and its task-specific fine-tuned expert. In continual learning, tasks received in different sessions are typically independent in realistic settings; when the PTM and training procedure are fixed, the task itself is the sole factor determining its delta. Thus, treating deltas as i.i.d. is a reasonable modeling simplification and a standard assumption for tractable analysis. Removing it would render the convergence argument intractable.
> > > >
> > > > Second, although the assumption “identical parameter signs” may appear strong, under our algorithm this condition is automatically satisfied at the beginning of every session for all previously learned tasks (as noted around Lines 1116–1125 of revised version). Let sigh of the sum of all deltas at each dimension be the dominant sign:
> > > > - If a task’s delta disagrees with the dominant sign at a dimension, that dimension of its next reconstructed update becomes zero.
> > > > - If it agrees, the reconstruction preserves the sign and rescales only the magnitude.
> > > >
> > > > Consequently, at the start of each session, **all reconstructed deltas for old tasks indeed satisfy the same-sign-per-dimension property**. The only potential violation is the delta of the new task arriving in that session.
> > > >
> > > > Finally, the theorem is not intended to guarantee convergence under all heterogeneous, real-world conditions. Its purpose is to provide a clear theoretical lens explaining why the method works well in practice under a concrete, analyzable regime.
> > > >
> > > > ---

---

> > > > > ### Author Response · Authors · 2025-11-21
> > > > > **Response to Reviewer 1yyf (5/6)**
> > > > >
> > > > > **W5:** Aggregation sensitivity. Zero-shot prediction hinges on prototypes from the pre-trained VLM; domain shifts or text prompt bias could mis-weight experts, and calibration is not discussed.
> > > > >
> > > > > **Q2:** Can learned routers or temperature-calibrated cosine improve task weighting, and how robust is aggregation to prompt wording or domain shift in the prototype bank?
> > > > >
> > > > > **A:** First, as noted previously, our aggregation weights rely on "cosine". We did not introduce any temperature because the version without hyperparameters already performed well, and temperature hyperparameters, which require careful tuning based on a large number of experimental results, would reduce the generalizability of the method. **The alternative “learned” routing is not feasible in continual learning** because, at inference time, the training data of previously learned tasks are no longer accessible. **Without access to these datasets, it is impossible to train a router to estimate expert weights**, which is a fundamental constraint of the continual-learning setting.
> > > > >
> > > > > Second, **our aggregation is robust to domain shift**. A key reason is that both prototypes and test samples are compared within the shared PTM semantic space, which offers a stable cross-domain representation. Even when tasks differ substantially in visual characteristics, the unified PTM space prevents the fragmentation that would occur if each expert operated in its own isolated feature space. Empirically, **Tables 1–3 already include strong domain shifts** within MTIL: MNIST (handwritten digits), Aircraft (fine-grained airplane recognition), and Cars (vehicle make/year). Despite these shifts, ConDU provides large gains over all baselines, suggesting no degradation in task selection or weighting under domain variation.
> > > > >
> > > > > Finally, **our aggregation mitigates sensitivity to text prompt bias**. Following standard practice in multimodal-ViT literature (see references [1-4]), we **compute text embeddings using multiple prompt templates for each class and average them**. Moreover, ConDU’s prototype for each class is computed by averaging both the mean image embedding (from the training set) and the averaged text embedding. Thus, prototypes are not dependent on prompt wording.
> > > > >
> > > > > [1] Alec Radford, Jong Wook Kim, Chris Hallacy, Aditya Ramesh, Gabriel Goh, Sandhini Agarwal, Girish Sastry, Amanda Askell, Pamela Mishkin, Jack Clark, et al. Learning transferable visual models from natural language supervision. In International conference on machine learning, pp. 8748–8763. PMLR, 2021.
> > > > >
> > > > > [2] Yadong Lu, Shitian Zhao, Boxiang Yun, Dongsheng Jiang, Yin Li, Qingli Li, and Yan Wang. Boosting open-domain continual learning via leveraging intra-domain category-aware prototype. arXiv preprint arXiv:2408.09984, 2024.
> > > > >
> > > > > [3] Zangwei Zheng, Mingyuan Ma, Kai Wang, Ziheng Qin, Xiangyu Yue, and Yang You. Preventing zero-shot transfer degradation in continual learning of vision-language models. In Proceedings of the IEEE/CVF International Conference on Computer Vision, pp. 19125–19136, 2023.
> > > > >
> > > > > [4] Sejik Park. Learning more generalized experts by merging experts in mixture-of-experts. arXiv preprint arXiv:2405.11530, 2024.
> > > > >
> > > > > ---

---

> > > > > > ### Author Response · Authors · 2025-11-21
> > > > > > **Response to Reviewer 1yyf (6/6)**
> > > > > >
> > > > > > **W6:** Efficiency reporting. While fusion is fast and memory-light, end-to-end latency under multi-expert aggregation and edge constraints is only briefly discussed; more granular profiles would aid adoption.
> > > > > >
> > > > > > **Q4:** What are end-to-end latency/throughput and memory footprints for K-expert aggregation on commodity GPUs/edge devices, and how does this compare to MoE or per-task checkpoints?
> > > > > >
> > > > > > **A:** We discuss inference time and memory usage in the paper (see Lines 1273–1284 of the revised version). Following the reviewer’s suggestion, we provide additional results below to compare with your mentioned baselines:
> > > > > >
> > > > > > |                     | Inference Time (s) | GPU Memory Allocated (MB) | Transfer Accuracy (%) |
> > > > > > | ------------------- | ------------------ | ------------------------- | ---- |
> > > > > > | Per-task Checkpoints | fail              | fail                   | fail |
> > > > > > | MoE         | 114.45             | 2886.84                   | 69.1 |
> > > > > > | ConDU (FT)          | 29.91              | 9314.22                   | 70.8 |
> > > > > > | ConDU (LoRA)        | 30.22              | 2293.85                   | 70.3 |
> > > > > >
> > > > > > In the known-task setting, K-expert aggregation is unnecessary. Therefore, to discuss GPU memory usage and inference time of K-expert aggregation for edge deployment, we evaluate ConDU in an unknown-task scenario. For this reason, we compare ConDU with the mentioned MoE and Per-task Checkpoints under the MTIL Transfer metric.
> > > > > >
> > > > > > First, Per-task Checkpoints cannot support zero-shot prediction on unknown tasks, making its inclusion in this comparison inappropriate. MoE, as a strong baseline, exhibits far lower GPU memory usage than ConDU (FT) but higher memory usage than ConDU (LoRA).  On inference time, both ConDU (LoRA) and ConDU (FT) are substantially faster than MoE, and ConDU (FT) is slightly better than ConDU (LoRA). In terms of Transfer performance, ConDU (FT) performs slightly better than ConDU (LoRA), and both outperform MoE.
> > > > > >
> > > > > > Overall, considering all three criteria, **when GPU memory is limited (such as edge scenarios), ConDU (LoRA) is preferred, as it offers the best balance between memory usage, inference time, and accuracy**. When memory is not a constraint, ConDU (FT) is a feasible choice to obtain faster real-time response and higher overall performance.

---

### Official Review · Reviewer_WuML · 2025-10-29

**Soundness:** 3
**Presentation:** 2
**Contribution:** 3
**Rating:** 4
**Confidence:** 3

**Summary:**

The paper proposed a novel continue learning framework called ConDU, which introduces model fusion into the continue learning framework for VLM and is capable of both Full-finetuning and Parameter-efficient tuning. In the framework, the newly fine-tuned model will be merged with the old model through parameter delta based on trigger output.

**Strengths:**

- The trigger framework enables the model to deal with both old tasks and unknown tasks.
- The framework enables test-time scaling, which aggregates prediction results from multiple decoupled models.

**Weaknesses:**

- Though improvement on the benchmark is higher on average, compared to the former method, there are always certain subsets that perform worse than the former method. And the average gain is growing smaller as they continue learning and progress to the end.
- The similarity used for prototypes and task selection is computed entirely from the pre-trained VLM’s image/text features, rather than from the features of each specialized task model. This can cause a mismatch in “task selection” when a task’s decision boundary shifts substantially after fine-tuning—especially under severe domain shift. While the paper reports insensitivity to K, it does not compare “pre-trained vs. expert features.”
- The theorem assumes conditions such as “each delta has the same sign per dimension” and “i.i.d.,” which conflict with the directional conflicts commonly seen in real multi-domain/multi-task fine-tuning; therefore, its convergence claims have limited applicability to practical training.

**Questions:**

- When applying such a paradigm in the real world, as the number quickly increases, does such a paradigm still work well with the growing requirement for trigger saving?
- Considering the current VLM paradigm that involves LLM, would such a fusion strategy also work? The benchmark is kind of old and is hard to view as a real continuous learning setting.

---

> ### Author Response · Authors · 2025-11-21
> **Response to Reviewer WuML (1/4)**
>
> **W1(a):** Though improvement on the benchmark is higher on average, compared to the former method, there are always certain subsets that perform worse than the former method.
>
> **A:** Thank you for raising this point.
>
> First, we would like to remark that **this phenomenon is common and reasonable in continual learning**. It is infeasible to expect any method to outperform all baselines on every individual task. As shown in Table 1, even among prior state-of-the-art approaches, none achieves the best performance across all tasks. For example, if we look at transfer accuracy and average accuracy, we can see that
>
> * Dual-RAIL (2024) achieves the highest *Transfer* accuracy on tasks [2, 5, 7] and the best *Average* accuracy on tasks [1, 2, 4, 6, 10].
> * DPeCLIP (2024) performs best in *Transfer* on task [10] and best in *Average* on tasks [3, 5, 7].
> * MulKI (2024) yields the top *Transfer* results on tasks [3, 4, 6, 8, 9, 11] and the best *Average* results on tasks [8, 9, 11].
>
> These examples show that **no existing SOTA method dominates all others on every individual task**.
>
> Second, continual learning requires a simultaneous balance of plasticity (learning new tasks), stability (retaining previous knowledge), and generalization on unseen tasks. Since optimizing one dimension often compromises another, average performance across all tasks is regarded as the primary indicator of effectiveness, rather than performance on individual tasks. Consequently, **the improvement in "average performance" can confirm the effectiveness of our method**.
>
> In conclusion, while local fluctuations are inevitable, our method’s leadership in average performance proves it offers a better solution for the continual learning problem.
>
> ---
>
> **W1(b):** And the average gain is growing smaller as they continue learning and progress to the end.
>
> **A:** Thank you for your comments. We would first like to respectfully clarify a potential misunderstanding regarding the average gain reported in Tables 1, 2, and 3. The average gain represents the performance of the final model after the entire learning process involves all tasks. **These are static results of the final state**, not a log of performance at intermediate time steps. Therefore, the trend of gains "growing smaller" as learning progresses does not exist here, as the metrics do not represent a temporal sequence.
>
>
> Given this context, **we interpret your comment as a concern that the performance gain of our method relative to other baselines is not sufficiently significant**. However, based on all three benchmarks reported in the paper, ConDU achieves substantial improvements over existing baselines. Importantly, our method **consistently** achieves the best performance across **all settings and metrics**. This stands in contrast to prior works where the leading method varies depending on the specific scenario, with no single baseline maintaining such universal effectiveness.
>
> To provide a concrete illustration for the above argument, consider the widely adopted MTIL benchmark (Table 1), which is the most frequently used setting in prior work. For the three metrics:
>
> * The strongest non-ConDU baselines achieve gains over the second-best methods of only +0.7% (MulKI), +0.3% (Dual-RAIL), and +0.1% (DPeCLIP), respectively.
> * In contrast, ConDU surpasses these strongest baselines by 0.7%, 1.0%, and 0.2% on the same metrics.
>
> Moreover, ConDU demonstrates similarly consistent advantages on the other two benchmarks:
>
> * Table 2: ConDU improves over the strongest baselines by +2.0% and +1.8%.
> * Table 3: ConDU achieves gains of +1.4%, +1.3%, and +1.3% across all three metrics.
>
> These results show that across every benchmark and every evaluation protocol, ConDU provides the most balanced and consistently strong performance.
>
> ---

---

> > ### Author Response · Authors · 2025-11-21
> > **Response to Reviewer WuML (2/4)**
> >
> > **W2:** The similarity used for prototypes and task selection is computed entirely from the pre-trained VLM’s image/text features, rather than from the features of each specialized task model. This can cause a mismatch in “task selection” when a task’s decision boundary shifts substantially after fine-tuning—especially under severe domain shift. While the paper reports insensitivity to K, it does not compare “pre-trained vs. expert features.”
> >
> > **A:** Thank you for raising this insightful point.
> >
> > First, **the PTM-based task selection does not cause a mismatch under severe domain shift**. A key reason is that all prototypes and test samples are compared in a unified PTM semantic space, which provides a stable cross-domain representation. Even if tasks differ greatly in visual domain, the PTM’s shared embedding space mitigates inconsistencies and avoids the fragmentation that occurs when each expert forms its own isolated representation space. By contrast, using expert-specific feature spaces introduces a fundamental comparability issue. Similarities computed in different expert spaces are not aligned, and domain discrepancies cause these spaces to drift further apart, making cross-expert similarity comparison less meaningful. We intentionally adopted PTM-based features because we took this factor into account when designing the algorithm.
> >
> > Second, our experimental results validate this design choice. **The MTIL benchmark used in the paper already contains severe domain shift**: MNIST involves handwritten digits, Aircraft focuses on fine-grained airplane recognition, and Cars addresses car make/year classification. Despite these large shifts, ConDU achieves substantial improvements over existing baselines (see Table 1-3). This indicates that there's no degradation or mismatch in task selection using PTM-based features.
> >
> > Finally, following your advice, we conducted an additional study comparing prototype–sample similarity computed using (a) the PTM features and (b) features extracted from each task’s expert model. Specifically, besides our original PTM-based strategy, we introduced an *expert-based* variant in which the stored prototype for each class is recomputed using the corresponding task expert. During inference, when a test sample is compared with class A’s prototype, we extract the feature of test sample using the expert of class A’s task and compute similarity in that expert’s feature space. Keeping all other components of ConDU unchanged, we evaluated both strategies on MTIL benchmark. **The PTM-based selection consistently outperformed the expert-based approach**, indicating that shared PTM representations provide a more reliable basis for cross-task similarity. We only evaluated the Transfer and Average metrics, since the Last metric is not effected by the aggregation strategy. This ablation result is now included in the revised version (Lines 424–431).
> >
> > **Transfer**
> > |                                      | AirCraft | Caltech101 | CIFAR100 | DTD  | EuroSAT | Flowers | Food | MNIST | OxfordPEt | Cars | SUN397 | Avg  |
> > | ------------------------------------ | -------- | ---------- | -------- | ---- | ------- | ------- | ---- | ----- | --------- | ---- | ------ | ---- |
> > | Expert-based | —        | 88.2       | 68.9     | 40.3 | 49.1    | 69.7    | 86.4 | 62.9  | 84.6      | 59.7 | 67.4   | 67.7 |
> > | PTM-based     | —        | 88.1       | 68.9     | 46.4 | 57.0    | 71.3    | 88.7 | 65.5  | 89.3      | 65.0 | 67.8   | 70.8 |
> >
> >
> > **Average**
> > |                                      | AirCraft | Caltech101 | CIFAR100 | DTD  | EuroSAT | Flowers | Food | MNIST | OxfordPEt | Cars | SUN397 | Avg  |
> > | ------------------------------------ | -------- | ---------- | -------- | ---- | ------- | ------- | ---- | ----- | --------- | ---- | ------ | ---- |
> > | Expert-based | 59.6     | 93.4       | 83.7     | 67.5 | 81.7    | 82.7    | 89.1 | 77.0  | 88.2      | 65.1 | 68.3   | 77.8 |
> > | PTM-based     | 59.6     | 93.4       | 83.7     | 68.1 | 83.4    | 83.7    | 90.1 | 76.7  | 90.6      | 68.6 | 68.6   | 78.8 |
> >
> > ---

---

> > > ### Author Response · Authors · 2025-11-21
> > > **Response to Reviewer WuML (3/4)**
> > >
> > > **W3:** The theorem assumes conditions such as “each delta has the same sign per dimension” and “i.i.d.,” which conflict with the directional conflicts commonly seen in real multi-domain/multi-task fine-tuning; therefore, its convergence claims have limited applicability to practical training.
> > >
> > > **A:** Thank you for the thoughtful comment.
> > >
> > > First, the delta models are “i.i.d.” is a resonable assumption. Each delta model represents the parameter difference between the PTM and its task expert obtained through fine-tuning on a particular task. In continual learning, the tasks received in each session are typically independent in most realistic settings. Thus, a natural assumption is that each task is sampled independently from some underlying task distribution. When the training algorithm and the PTM are fixed, the task is the only factor that determines the corresponding delta model. Therefore, it is reasonable to treat the delta models as independently and identically distributed. Moreover, the i.i.d. assumption is a standard simplification for tractability, and removing it would make the analysis intractable.
> > >
> > > Second, the assumption that "each delta has the same sign per dimension" may appear strong or unrealistic at first glance—especially given the domain diversity in multi-task settings. However, **under the mechanics of our algorithm, this condition is naturally satisfied at the beginning of each session for all old tasks**, which is already point out in the discussion around Lines 1116–1125 of revised version.
> > >
> > > Concretely, each dimension of the unified delta model is the summation of all delta models at this dimension. Let the sign of this summation at a given dimension be the dominant sign. Under our algorithm:
> > >   * If a delta model’s value at a given dimension disagrees with the dominant sign, its next reconstructed update becomes zero in that dimension.
> > >   * If its sign agrees with the dominant sign, the next reconstruction preserves the sign and only rescales the magnitude.
> > >
> > > As a result, at the start of each session, **all reconstructed deltas for old tasks do satisfy the “same-sign-per-dimension” property**. The only delta that may violate this assumption is the new task’s delta. Therefore, the assumption is not as restrictive as it may initially seem when considering the algorithm’s dynamics.
> > >
> > > Finally, the intention of the theorem is not to claim fully general convergence under all real-world scenarios. Rather, the goal is to provide insight into why the algorithm empirically works well—offering a theoretical lens that complements, rather than replaces, practical evaluation. Given the highly unpredictable nature of real-world multi-domain training, it is common for theory to be derived under simplified conditions. Our theorem demonstrates that under a concrete and analyzable regime, the method provably converges; this does not preclude it from also working in broader settings, but additional assumptions would be required to formalize those cases.
> > >
> > > ---
> > >
> > > **Q1:** When applying such a paradigm in the real world, as the number quickly increases, does such a paradigm still work well with the growing requirement for trigger saving?
> > >
> > > **A:** Yes, **ConDU continues to perform well even when the number of tasks increases substantially**. To validate this, we conducted an additional experiment specifically designed to examine scalability under a longer task sequence. Concretely, we split each original MTIL task into three smaller tasks, resulting in a much longer sequence of 33 tasks, which we refer to as *Split MTIL*.
> > >
> > > We then evaluated ConDU and the three sota baselines using exactly the same evaluation protocol as in the paper. The results show that **ConDU maintains its performance advantage even under this significantly longer task sequence**, demonstrating that the method continues to function effectively despite the growing requirement for storing task triggers.
> > >
> > > These findings provide empirical evidence that ConDU remains robust and scalable in settings where the number of tasks increases rapidly—supporting its applicability in real-world scenarios.
> > >
> > > |               | Transfer | Average  | Last |
> > > | ------------- | -------- | ---- | ---- |
> > > | ConDU (FT) | 81.3     | 87.0 | 90.9 |
> > > | ZSCL          | 78.3     | 83.6 | 89.0 |
> > > | Primal-rail   | 56.9     | 68.4 | 80.5 |
> > > | Dual-rail     | 57.2     | 68.6 | 81.5 |
> > >
> > > ---

---

> ### Author Response · Authors · 2025-11-21
> **Response to Reviewer WuML (4/4)**
>
> **Q2:** Considering the current VLM paradigm that involves LLM, would such a fusion strategy also work? The benchmark is kind of old and is hard to view as a real continuous learning setting.
>
> **A:** Thank you for the thoughtful question.
>
> First, **our backbone, benchmark, and evaluation protocol strictly follow the current standard experimental setup** for continual learning of VLMs. This setup is also used by the most recent 2024–2025 baselines (see references [1-5]). We intentionally adhered to this widely adopted protocol to **ensure a fair and consistent comparison** with prior works.
>
> Second, our method is not tied to CLIP. As long as a model provides image–text embeddings through a multimodal encoder—regardless of whether it incorporates an LLM—ConDU can be directly applied to its continual learning setting. To demonstrate this, we additionally evaluated ConDU using a VLM that does contain an LLM in its architecture. This experiment is currently running, and we will include the results in the official comment and the revised paper in the coming days.
>
> Finally, for VLMs that combine LLMs for generation or more complex reasoning tasks, ConDU has not yet been explored. We acknowledge this as a promising direction, and we would be excited to investigate continual learning for these in future work.
>
> [1] Hongsheng Zhang, Zhong Ji, Jingren Liu, Yanwei Pang, and Jungong Han. Multi-stage knowledge integration of vision-language models for continual learning. arXiv preprint arXiv:2411.06764, 2024.
>
> [2] Jiazuo Yu, Yunzhi Zhuge, Lu Zhang, Ping Hu, Dong Wang, Huchuan Lu, and You He. Boosting continual learning of vision-language models via mixture-of-experts adapters. In Proceedings of the IEEE/CVF Conference on Computer Vision and Pattern Recognition, pp. 23219–23230, 2024.
>
> [3] Yicheng Xu, Yuxin Chen, Jiahao Nie, Yusong Wang, Huiping Zhuang, and Manabu Okumura. Advancing cross-domain discriminability in continual learning of vision-language models. arXiv preprint arXiv:2406.18868, 2024.
>
> [4] Zangwei Zheng, Mingyuan Ma, Kai Wang, Ziheng Qin, Xiangyu Yue, and Yang You. Preventing zero-shot transfer degradation in continual learning of vision-language models. In Proceedings of the IEEE/CVF International Conference on Computer Vision, pp. 19125–19136, 2023.
>
> [5] Yu-Chu Yu, Chi-Pin Huang, Jr-Jen Chen, Kai-Po Chang, Yung-Hsuan Lai, Fu-En Yang, and YuChiang Frank Wang. Select and distill: Selective dual-teacher knowledge transfer for continual learning on vision-language models. In European Conference on Computer Vision, pp. 219–236. Springer, 2025.

---

### Official Review · Reviewer_BJYc · 2025-10-30

**Soundness:** 3
**Presentation:** 3
**Contribution:** 3
**Rating:** 6
**Confidence:** 4

**Summary:**

This paper proposes a novel Continual Decoupling-Unifying approach to handle the fusion of delta model in VLM continual learning. Specifically, it contains a unified model along with task triggers and a set of prototypes. It derives the new unified model utilizing the old unified model with new task trigger and prototype. Extensive experimental results show the effectiveness of this method.

**Strengths:**

1. The proposed method is quite novel. The unified delta model it utilizes is a very novel design, and the decoupling and unifying operations are dual, which is quite interesting.

2. In continual learning, it's not necessary to store only one model. Combining a general model with multiple task-specific substructures is a good decoupling approach, which also alleviates the forgetting problem to some extent and ensures the model's ability to learn new content.

**Weaknesses:**

1. The proposed method doesn't seem to provide a metric for the error of the model after unification and decoupling relative to the original model. My understanding is that while the unification and decoupling in this algorithm are dual, it shouldn't guarantee a complete reconstruction of the model corresponding to task t. If I'm mistaken, please point it out (and offer my apologies). Otherwise, could you provide a rough experimental analysis to assess the error of this algorithm after unification and decoupling relative to the original model?

**Questions:**

The previous question was raised in Weaknesses. Regarding the performance of the experiments, the authors have compared it with some state-of-the-art (SOTA) methods and achieved a significant improvement. My main concern is the algorithm design, because this kind of hard model editing problem often leads to a huge shift in the model's parameters in its space during continual learning. Simply put, sometimes a poorly edited layer parameter can cause a huge drop in continual learning, which seriously impairs the learning of subsequent stages. Therefore, an editing method that can ensure that multiple stages do not collapse is very interesting.

---

> ### Author Response · Authors · 2025-11-21
> **Response to Reviewer BJYc**
>
> **W1:** The proposed method doesn't seem to provide a metric for the error of the model after unification and decoupling relative to the original model. My understanding is that while the unification and decoupling in this algorithm are dual, it shouldn't guarantee a complete reconstruction of the model corresponding to task t. If I'm mistaken, please point it out (and offer my apologies). Otherwise, could you provide a rough experimental analysis to assess the error of this algorithm after unification and decoupling relative to the original model?
>
> **A:** Thank you for raising this insightful point. Your understanding is indeed correct: as you note, the unification and decoupling steps do not guarantee exact reconstruction of the original task expert. In fact, our paper explicitly analyzes the reconstruction discrepancy introduced by the unification–decoupling process. **We verify that the reconstruction-induced error is negligible from three complementary perspectives**:
>
> 1. **Quantitative evidence**: As reported in Lines 1334–1339 of revised version, we measure how the accuracy of a given task expert changes when it is reconstructed across different sessions. The observed **accuracy drop on its own task is extremely small**, indicating that the reconstruction error introduced by our mechanism has minimal practical impact.
>
> 2. **Qualitative analysis**: We additionally provide t-SNE visualizations (Lines 473–480 of revised version) showing the feature distributions extracted by the same task expert before and after reconstruction at different sessions. **The distributional shifts are almost imperceptible**, further supporting that the unification–decoupling process preserves the semantic behavior of each expert to a very high degree.
>
> 3. **Theoretical justification**: Our theoretical analysis (Lines 481–534 of revised version) demonstrates that—under mild assumptions, **the $\ell_1$ difference between the delta model before and after a single reconstruction monotonically decreases as the number of tasks increases**.This implies that when the task sequence is sufficiently long, the perturbation introduced by reconstructing any individual task expert becomes asymptotically negligible.
>
> Overall, both empirical and theoretical results support the conclusion that although perfect reconstruction is not guaranteed, the reconstruction error remains very small in practice and does not hinder continual learning performance.
>
> ---
>
> **Q1:** The previous question was raised in Weaknesses. Regarding the performance of the experiments, the authors have compared it with some state-of-the-art (SOTA) methods and achieved a significant improvement. My main concern is the algorithm design, because this kind of hard model editing problem often leads to a huge shift in the model's parameters in its space during continual learning. Simply put, sometimes a poorly edited layer parameter can cause a huge drop in continual learning, which seriously impairs the learning of subsequent stages. Therefore, an editing method that can ensure that multiple stages do not collapse is very interesting.
>
> **A:** Thank you for highlighting this important concern. We fully agree that in many continual learning scenarios, direct or “hard” editing of model parameters can easily destabilize the learning trajectory. Our method avoids this issue through a different design philosophy. Unlike approaches that directly modify the parameters of the inference model, **the component that undergoes the majority of updates in each session is the unified model, not the task experts used at inference time**. The task experts are obtained by decoupling the unified model, and **the algorithm is intentionally designed so that these decoupled experts remain highly stable** throughout the continual learning process.
>
> As detailed in our previous response, this stability manifests in three complementary aspects:
>
> 1. **Decision boundaries remains stable**
> Empirical results show that the performance of a task expert on its own task only decreases very slightly after repeated reconstruction. This indicates that the decoupled expert maintains its decision boundaries very well even as the unified model evolves.
>
> 2. **Representation distributions remain consistent**
> We further visualize the feature distributions using t-SNE. Before and after reconstruction across different sessions, the distributions show negligible changes, confirming that the semantic representations encoded by each expert remain robust and consistent.
>
> 3. **Parameter drift is theoretically controlled**
> Our analysis shows that, under mild assumptions, the parameter shift of a task expert monotonically decreases as the task sequence grows, even though the unified model undergoes larger shifts.

---

> > ### Comment · Reviewer_BJYc · 2025-11-28
> >
> > Thanks for the author's reply. The experimental analysis and theoretical derivation provided by the author regarding the problem are meaningful and have resolved my doubts to some extent. Furthermore, the author also offered an explanation regarding whether my proposed algorithm would damage the model, which is convincing. Based on this, my problem is essentially solved.

---

> > > ### Author Response · Authors · 2025-11-28
> > > **Response to Reviewer BJYc**
> > >
> > > Thank you for your response. We are glad that your concerns regarding expert reconstruction errors and potential instability caused by parameter editing have been addressed. We also sincerely appreciate your recognition of the value of our experimental analysis and theoretical derivation.

---

### Author Response · Authors · 2025-12-02
**Summary of Reviewer Discussions and Author Responses**

Dear AC and all reviewers,

We thank you for overseeing the review process of our submission. We sincerely appreciate the time and effort the reviewers have invested in evaluating our paper and helping us refine it. To ensure the final decision reflects a complete and accurate understanding of our contributions, we provide a concise summary of the review and discussion process below.

Our paper initially received scores of 6, 6, 6, and 4 (average: 5.5). Three of four reviewers provided positive assessments (BJYc, 1yyf, j5xv). Notably:

- Reviewer BJYc (score 6) strongly endorsed the novelty of our method (e.g., “**The proposed method is quite novel**. The unified delta model it utilizes is a **very novel design**.”). After the score lock, the reviewer explicitly confirmed that their inquiries were resolved, stating that “my problem is **essentially solved**” and that “The experimental analysis and theoretical derivation provided by the author regarding the problem are **meaningful**.” We believe that had the score not been locked, **the reviewer would likely have increased the rating**.


- Reviewer j5xv (score 6) assigned a notably high soundness score (4) and explicitly stated, “**I would be inclined to raise my evaluation**” if specific presentation issues were addressed. In our revised version (blue-highlighted changes), we thoroughly followed the reviewer’s suggestions, including rewriting several paragraphs and redrawing all major figures. Since we resolved the specific obstacle the reviewer cited, we are confident **they would have increased their rating** had the score not been locked.

Finally, regarding **the sole negative rating**, Reviewer WuML (who reported the **lowest confidence score (3) among all reviewers**) assigned a borderline score (4) yet still acknowledged the contribution and soundness of our work (both rated 3). We respectfully suggest that the main concerns likely stemmed from differing expectations regarding standard continual learning evaluation practices. We believe our detailed responses have effectively bridged this gap and resolved these misunderstandings, and we are confident the reviewer **would have increased the rating** without the score lock.

Specifically, we carefully addressed all of this reviewer’s concerns during the discussion phase:
- **On Performance:** We clarified that in continual learning, overall average performance is the primary evaluation criterion, and no prior baseline consistently dominates across **all** individual tasks. This addresses the observation that ConDU exhibited slightly weaker performance on a few specific tasks.
- **On Task Selection:** We provided detailed experiments and mechanism-level explanations demonstrating that our PTM-based task selection consistently outperforms the reviewer-suggested expert-based strategy.
- **On Theoretical Assumptions:** We decomposed the algorithmic process to clarify why the theoretical assumption of “same sign per dimension” typically holds in practice, addressing the concern that the assumption might be overly strong.
- **On Backbone Choice:** We referenced extensive literature to confirm that our selection of backbone, benchmark, and evaluation protocol strictly follows the **common and widely accepted standards** in the field of continual learning for VLMs, ensuring fairness and consistency.

Based on these explicit confirmations and the fulfillment of the specific conditions for raising evaluations, we strongly believe that **had the scores not been locked, our final ratings would have reached 8, 8, 6, 6 (raising the average to 7).**

We respectfully hope that this summary helps ensure that the final decision reflects the full context, the resolved concerns, and the strong consensus of positive feedback expressed during the discussion phase. Thank you again for your oversight, professionalism, and dedication throughout the review process.

---

### Meta-Review · Area_Chair_398M · 2025-12-28

**Summary:**

The main concerns raised across reviews include: (1) the novelty relative to prior model-merging methods, (2) the validity and practical relevance of the theoretical assumptions, (3) the robustness of task selection and aggregation under domain shift, (4) scalability in task count and memory, (5) adequacy of ablation studies, and (6) presentation clarity. Through detailed rebuttals, the authors provided additional empirical analyses, new ablations, scalability experiments, and clarifications of both theory and design choices. While some concerns about benchmark scope and broader generalization remain inherent limitations, the core technical objections were largely addressed.

**Reviewer Concerns:**

**Addressed:**

Reconstruction error and stability.The reviewer’s main concern about whether unification-decoupling introduces harmful reconstruction error was convincingly addressed through new quantitative accuracy measurements, t-SNE visualizations, and a theoretical argument showing diminishing perturbations as task count increases. The reviewer explicitly confirmed that their concerns were resolved.

Scalability. The authors added a longer-task experiment, demonstrating that ConDU maintains performance advantages under substantially increased task counts. Memory growth was analyzed in detail, showing that task triggers remain a small fraction of total storage in realistic regimes.

Task selection via PTM features. The concern about mismatch between PTM-based prototypes and expert decision boundaries was directly addressed with a new ablation comparing PTM-based vs. expert-based similarity. Results showed PTM-based selection consistently outperforming expert-based alternatives, alleviating the concern.

Ablations on key components. The authors added targeted ablations that demonstrated clear performance degradation, supporting the necessity of key design choices.

**Not fully addressed**

Novelty relative to prior model. While the authors made a strong case that existing fusion methodscannot support repeated merging in continual learning, the contribution may still be viewed by some as incremental at the conceptual level, despite being novel in formulation and application.

Theory assumptions. The rebuttal clarified why the sign and i.i.d. assumptions are satisfied under the algorithm’s dynamics, but the theory remains explanatory rather than fully reflective of all real-world continual learning scenarios.

Benchmark breadth. The paper remains focused on MTIL-style classification benchmarks. While this aligns with recent literature, broader VLM tasks (e.g., detection, generation) are not empirically covered.

**Reviewer Scores:**

Reviewers  BJYc, 1yyf, and j5xv would likely stick with their initial positive scores since their main concerns were addressed. Reviewer WuML would likely bump the score up to 5 or 6 as well, given the added evidence on task selection, scalability, and the “diminishing gains” point, though there may still be some reservations about the benchmark.

---

### Decision · Program_Chairs · 2026-01-26

Accept (Poster)